# Towards Unraveling and Improving Generalization in World Models

## Abstract

World models have recently emerged as a promising approach for reinforcement learning (RL), as evidenced by its stimulating successes that world model based agents achieve state-of-the-art performance on a wide range of tasks in empirical studies. The primary goal of this study is to obtain a deep understanding of the mysterious generalization capability of world models, based on which we devise new methods to enhance it further. Thus motivated, we develop a stochastic differential equation formulation by treating the world model learning as a stochastic dynamic system in the latent state space, and characterize the impact of latent representation errors on generalization, for both cases with zero-drift representation errors and with non-zero-drift representation errors. Our somewhat surprising findings, based on both theoretic and experimental studies, reveal that for the case with zero drift, modest latent representation errors can in fact function as implicit regularization and hence result in generalization gain. We further propose a Jacobian regularization scheme to mitigate the compounding error propagation effects of non-zero drift, thereby enhancing training stability and generalization. Our experimental results corroborate that this regularization approach not only stabilizes training but also accelerates convergence and improves performance on predictive rollouts.

## 1 Introduction

Model-based reinforcement learning (RL) has emerged as a promising learning paradigm to improve sample efficiency by enabling agents to exploit a learned model for the physical environment. Notably, in recent works [14, 13, 15, 16, 21, 10, 32, 22] on world models, an RL agent learns the latent dynamics model of the environment, based on the observations and action signals, and then optimizes the policy over the learned dynamics model. Different from conventional approaches, world-model based RL takes an *end-to-end learning* approach, where the building blocks (such as dynamics model, perception and action policy) are trained and optimized to achieve a single overarching goal, offering significant potential to improve generalization capability. For example, DreamerV2 and DreamerV3 achieve great progress in mastering diverse tasks involving continuous and discrete actions, image-based inputs, and both 2D and 3D environments, thereby facilitating robust learning across unseen task domains [14, 13, 15]. Recent empirical studies have also demonstrated the capacity of world models to generalize to unseen states in complex environments, such as autonomous driving [19]. Nevertheless, it remains not well understood when and how world models can generalize well in unseen environments.

In this work, we aim to first obtain a deep understanding of the *generalization* capability of world models by examining the impact of *latent representation errors*, and then to devise new methods to enhance its generalization. While one may expect that optimizing a latent dynamics model (LDM) prior to training the task policy would minimize latent representation errors and hence can achieve better world model training, our somewhat surprising findings, based on both theoretical and empirical

| batch size \ perturbation | $\alpha = 10$ | $\alpha = 20$ | $\alpha = 30$ | $\beta = 25$ | $\beta = 50$ | $\beta = 75$ |
|---|---|---|---|---|---|---|
| 8 | 691.62 | 363.73 | 153.67 | 624.67 | 365.31 | 216.52 |
| 16 | **830.39** | 429.62 | **213.78** | **842.26** | **569.42** | **375.61** |
| 32 | **869.39** | **436.87** | **312.99** | **912.12** | **776.86** | **655.26** |
| 64 | 754.47 | **440.44** | 80.24 | 590.41 | 255.2 | 119.62 |

Table 1: Reward values on unseen perturbed states by rotation ($\alpha$) or mask ($\beta\%$) with $\mathcal{N}(0.15, 0.5)$.

studies, reveal that modest latent representation errors in the training phase may in fact be beneficial. In particular, the alternating training strategy for world model learning, which simultaneously refines both the LDM and the action policy, could actually bring generalization gain, because the modest latent representation errors (and the corresponding induced gradient estimation errors) could enable the world model to visit unseen states and thus lead to improved generalization capacities. For instance, as shown in Table 1, our experimental results suggest that moderate batch sizes (e.g., 16 or 32) appear to position the induced errors within a regime conferring notable generalization benefits, leading to higher generalization improvement, when compared to the cases with very small (e.g., 8) or large (e.g., 64) batch sizes.

In a nutshell, *latent representation errors* incurred by latent encoders, if designed properly, may actually facilitate world model training and enhance generalization. This insight aligns with recent advances in deep learning, where noise injection schemes have been studied as a form of implicit regularization to enhance models' robustness. For instance, recent study [2] analyzes the effects of introducing isotropic Gaussian noise at each layer of neural networks, identifying it as a form of implicit regularization. Another recent work [27] explores the addition of zero-drift Brownian motion to RNN architectures, demonstrating its regularizing effects in improving network's stability against noise perturbations.

We caution that *latent representation errors* in world models differ from the above noise injection schemes ([27, 2]), in the following aspects: 1) Unlike the artificially injected noise only added in training, these errors are inherent in world models, leading to error propagation in the rollouts; 2) Unlike the controlled conditions of isotropic or zero-drift noise examined in prior studies, the errors in world models may not exhibit such well-behaved properties in the sense that the drift may be non-zero and hence biased; 3) additionally, in the iterative training of world models and agents, the error originating from the encoder affects the policy learning and agent exploration. In light of these observations, we develop a continuous-time stochastic differential equation (SDE) formulation by treating the world model learning as a stochastic dynamic system with stochastic latent states. This approach offers an insightful view on model errors as stochastic perturbation, enabling us to obtain an explicit characterization to quantify the impacts of the errors on world models' generalization capability. Our main contributions can be summarized as follows.

- *Latent representation errors as implicit regularization:* Aiming to understand the generalization capability of world models and improve it further, we develop a continuous-time SDE formulation by treating the world model learning as a stochastic dynamic system in latent state space. Leveraging tools in stochastic calculus and differential geometry, we characterize the impact of latent representation errors on world models' generalization. Our findings reveal that under some technical conditions, modest latent representation errors can in fact function as implicit regularization and hence result in generalization gain.

- *Improving generalization in non-zero drift cases via Jacobian regularization:* For the case where latent representation errors exhibit non-zero drifts, we show that the additional bias term would degrade the implicit regulation and hence may make the learning unstable. We propose to add Jacobian regularization to mitigate the effects of non-zero-drift errors in training. Experimental studies are carried out to evaluate the efficacy of Jacobian regularization.

- *Reducing error propagation in predictive rollouts:* We explicitly characterize the effect of latent representation errors on predictive rollouts. Our experimental results corroborate that Jacobian regularization can reduce the impact of error propagation on rollouts, leading to enhanced prediction performance and accelerated convergence in tasks with longer time horizons.

- *Bounding Latent Representation Error:* We establish a novel bound on the latent representation error within CNN encoder-decoder architectures. To our knowledge, this is the first quantifiable

bound applied to a learned latent representation model, and the analysis carries over to other architectures (e.g., ReLU) along the same line.

**Notation.** We use Einstein summation convention for succinctness, where $a_i b_i$ denotes $\sum_i a_i b_i$. We denote functions in $\mathcal{C}^{k,\alpha}$ as being $k$-times differentiable with $\alpha$-Hölder continuity. The Euclidean norm of a vector is represented by $\|\cdot\|$, and the Frobenius norm of a matrix by $|\cdot|_F$; this notation may occasionally extend to tensors. The notation $x^i$ indicates the $i^{th}$ coordinate of the vector $x$, and $A^{ij}$ the $(i, j)$-entry of the matrix $A$. Function composition is denoted by $f \circ g$, implying $f(g)$. For a differentiable function $f : \mathbb{R}^n \to \mathbb{R}^m$, its Jacobian matrix is denoted by $\frac{\partial f}{\partial x} \in \mathbb{R}^{m \times n}$. Its gradient, following conventional definitions, is denoted by $\nabla f$. The constant $C$ may represent different values in distinct contexts.

## 2 Related Work

**World model based RL.** World models have demonstrated remarkable efficacy in visual control tasks across various platforms, including Atari [1] and Minecraft [8], as detailed in the studies by Hafner et al. [14, 13, 15]. These models typically integrate encoders and memory-augmented neural networks, such as RNNs [33], to manage the latent dynamics. The use of variational autoencoders (VAE) [7, 23] to map sensory inputs to a compact latent space was pioneered by Ha et al. [12]. Furthermore, the Dreamer algorithm [13, 16] employs convolutional neural networks (CNNs) [24] to enhance the processing of both hidden states and image embeddings, yielding models with improved predictive capabilities in dynamic environments.

**Continuous-time RNNs.** The continuous-time assumption is standard for theoretical formulations of RNN models. Li et al. [26] study the optimization dynamics of linear RNNs on memory decay. Chang et al. [4] propose AntisymmetricRNN, which captures long-term dependencies through the control of eigenvalues in its underlying ODE. Chen et al. [5] propose the symplectic RNN to model Hamiltonians. As continuous-time formulations can be discretized with Euler methods [4, 5] (or with Euler-Maruyama methods if stochastic in [27]) and yield similar insights, this step is often eliminated for brevity.

**Implicit regularization by noise injection in RNN.** Studies on noise injection as a form of implicit regularization have gained traction, with Lim et al. [27] deriving an explicit regularizer under small noise conditions, demonstrating bias towards models with larger margins and more stable dynamics. Camuto et al. [2] examine Gaussian noise injections at each layer of neural networks. Similarly, Wei et al. [31] provide analytic insights into the dual effects of dropout techniques.

## 3 Demystifying World Model: A Stochastic Differential Equation Approach

As pointed out in [14, 13, 15, 16], critical to the effectiveness of the world model representation is the stochastic design of its latent dynamics model. The model can be outlined by the following key components: an encoder that compresses high dimensional observations $s_t$ into a low-dimensional latent state $z_t$ (Eq.1), a sequence model that captures temporal dependencies in the environment (Eq.2), a transition predictor that estimates the next latent state (Eq.3), and a latent decoder that reconstructs observed information from the posterior (Eq.4):

$$\text{Latent Encoder: } z_t \sim q_{\text{enc}}(z_t \,|\, h_t, s_t), \tag{1}$$

$$\text{Sequence Model: } h_t = f(h_{t-1}, z_{t-1}, a_{t-1}), \tag{2}$$

$$\text{Transition Predictor: } \tilde{z}_t \sim p(\tilde{z}_t \,|\, h_t), \tag{3}$$

$$\text{Latent Decoder: } \tilde{s}_t \sim q_{\text{dec}}(\tilde{s}_t \,|\, h_t, \tilde{z}_t) \tag{4}$$

In this work, we consider a popular class of world models, including Dreamer and PlaNet, where $\{z, \tilde{z}, \tilde{s}\}$ have distributions parameterized by neural networks' outputs, and are Gaussian when the outputs are known. It is worth noting that $\{z, \tilde{z}, \tilde{s}\}$ may not be Gaussian and are non-Gaussian in general. This is because while $z$ is conditional Gaussian, its mean and variance are random variables which are learned by the encoder with $s$ and $h$ being the inputs, rendering that $z$ is non-Gaussian due to the mixture effect. For this setting, we have a continuous-time formulation where the latent dynamics model can be interpreted as stochastic differential equations (SDEs) with coefficient functions of known inputs. Due to space limitation, we refer to Proposition B.1 in the Appendix for a more detailed treatment.

Consider a complete, filtered probability space $(\Omega, \mathcal{F}, \{\mathcal{F}_t\}_{t\in[0,T]}, \mathbb{P})$ where independent standard Brownian motions $B_t^{\text{enc}}, B_t^{\text{pred}}, B_t^{\text{seq}}, B_t^{\text{dec}}$ are defined such that $\mathcal{F}_t$ is their augmented filtration, and $T \in \mathbb{R}$ as the time length of the task environment. We interpret the stochastic dynamics of LDM with latent representation errors through coupled SDEs representing continuous-time analogs of the discrete components:

$$\text{Latent Encoder: } d\,z_t = (q_{\text{enc}}(h_t, s_t) + \varepsilon\,\sigma(h_t, s_t))\,dt + (\bar{q}_{\text{enc}}(h_t, s_t) + \varepsilon\,\bar{\sigma}(h_t, s_t))\,dB_t^{\text{enc}}, \quad (5)$$

$$\text{Sequence Model: } d\,h_t = f(h_t, z_t, \pi(h_t, z_t))\,dt + \bar{f}(h_t, z_t, \pi(h_t, z_t))\,dB_t^{\text{seq}} \quad (6)$$

$$\text{Transition Predictor: } d\,\tilde{z}_t = p(h_t)\,dt + \bar{p}(h_t)\,dB_t^{\text{pred}}, \quad (7)$$

$$\text{Latent Decoder: } d\,\tilde{s}_t = q_{\text{dec}}(h_t, \tilde{z}_t)\,dt + \bar{q}_{\text{dec}}(h_t, \tilde{z}_t)\,dB_t^{\text{dec}}, \quad (8)$$

where $\pi(h, \tilde{z})$ is a policy function as a local maximizer of value function and the stochastic process $s_t$ is $\mathcal{F}_t$-adapted. Notice that $\bar{f}$ is often a zero function indicating that Equation (6) is an ODE, as the sequence model is generally designed as deterministic. Generally, the coefficient functions in $dt$ and $dB_t$ terms in SDEs are referred to as the *drift* and *diffusion* coefficients. Intuitively, the diffusion coefficients here represent the stochastic model components. In Equation (5), $\sigma(\cdot, \cdot)$ and $\bar{\sigma}(\cdot, \cdot)$ denotes the drift and diffusion coefficients of the *latent representation errors*, respectively. Both are assumed to be functions of hidden states $h_t$ and task states $s_t$. In addition, $\varepsilon$ indicates the magnitude of the error.

Next, we impose standard assumptions on these SDEs (5) - (8) to guarantee the well-definedness of the solution to SDEs. For further technical details, we refer readers to fundamental works on SDEs in the literature (e.g.,[30, 17]).

**Assumption 3.1.** The drift coefficient functions $q_{\text{enc}}, f, p$ and $q_{\text{dec}}$ and the diffusion coefficient functions $\bar{q}_{\text{enc}}, \bar{p}$ and $\bar{q}_{\text{dec}}$ are bounded and Borel-measurable over the interval $[0, T]$, and of class $\mathcal{C}^3$ with bounded Lipschitz continuous partial derivatives. The initial values $z_0, h_0, \tilde{z}_0, \tilde{s}_0$ are square-integrable random variables.

**Assumption 3.2.** $\sigma$ and $\bar{\sigma}$ are bounded and Borel-measurable and are of class $\mathcal{C}^3$ with bounded Lipschitz continuous partial derivatives over the interval $[0, T]$.

## 3.1 Latent Representation Errors in CNN Encoder-Decoder Networks

As shown in the empirical studies with different batch sizes (Table 1), the latent representation error would also enrich generalization when it is within a moderate regime. In this section, we show that the latent representation error, in the form of approximation error corresponding to widely used CNN encoder-decoder, could be made sufficiently small by finding appropriate CNN network configuration. In particular, this result provides theoretical justification to interpreting latent representation error as stochastic perturbation in the dynamical system defined in Equations (5 - 8), as the error magnitude $\varepsilon$ can be made sufficiently small by CNN network configuration.

Consider the state space $\mathcal{S} \subset \mathbb{R}^{d_{\mathcal{S}}}$ and the latent space $\mathcal{Z}$. Consider a state probability measure $Q$ on the state space $\mathcal{S}$ and a probability measure $P$ on the latent space $\mathcal{Z}$. As high-dimensional state space in image-based tasks frequently exhibit *intrinsic lower-dimensional geometric structure*, we adopt the latent manifold assumption, formally stated as follows:

**Assumption 3.3.** (Latent manifold assumption) For a positive integer $k$, there exists a $d_{\mathcal{M}}$-dimensional $\mathcal{C}^{k,\alpha}$ submanifold $\mathcal{M}$ (with $\mathcal{C}^{k+3,\alpha}$ boundary) with Riemannian metric $g$ and has positive reach and also isometrically embedded in the state space $\mathcal{S} \subset \mathbb{R}^{d_{\mathcal{S}}}$ and $d_{\mathcal{M}} \ll d_{\mathcal{S}}$, where the state probability measure is supported on. In addition, $\mathcal{M}$ is a compact, orientable, connected manifold.

**Assumption 3.4.** (Smoothness of state probability measure) $Q$ is a probability measure supported on $\mathcal{M}$ with its Radon-Nikodym derivative $q \in \mathcal{C}^{k,\alpha}(\mathcal{M}, \mathbb{R})$ w.r.t $\mu_{\mathcal{M}}$.

Let $\mathcal{Z}$ be a closed ball in $\mathbb{R}^{d_{\mathcal{M}}}$, that is $\{x \in \mathbb{R}^{d_{\mathcal{M}}} : \|x\| \leq 1\}$. $P$ is a probability measure supported on $\mathcal{Z}$ with its Radon-Nikodym derivative $p \in \mathcal{C}^{k,\alpha}(\mathcal{Z}, \mathbb{R})$ w.r.t $\mu_{\mathcal{Z}}$. In practice, it is usually an easy-to-sample distribution such as uniform distribution which is determined by a specific encoder-decoder architecture choice.

**Latent Representation Learning**. We define the *latent representation learning* as to find encoder $g_{\text{enc}} : \mathcal{M} \to \mathcal{Z}$ and decoder $g_{\text{dec}} : \mathcal{Z} \to \mathcal{M}$ as maps that optimize the following objectives:

$$\min_{g_{\text{enc}} \in \mathcal{G}} W_1\left(g_{\text{enc}\#}\,Q,\,P\right); \qquad \min_{g_{\text{dec}} \in \mathcal{G}} W_1\left(Q,\,g_{\text{dec}\#}\,P\right).$$

Here, $g_{\text{enc}_\#} Q$ and $g_{\text{dec}_\#} P$ represent the pushforward measures of $Q$ and $P$ through the encoder map $g_{\text{enc}}$ and decoder map $g_{\text{dec}}$, respectively. The latent representation error is understood as the "difference" of pushforward measure by the encoder/decoder and target measure. Here, *to understand the "scale" of the error $\varepsilon$ in Equation (5), we use $W_1$ for the discrepancy between probability measures.* In particular, for Dreamer-type loss function that uses KL-divergence, we note that squared $W_1$ distance between two probability measures can be upper bounded by their KL-divergence up to a constant [11], implying that one could reasonably expect the $W_1$ distance to also decrease when KL-divergence is used in the model.

**CNN configuration.** As a popular choice choice in encoder-decoder architecture is CNN, we consider a general CNN function $f_{\text{CNN}} : \mathcal{X} \to \mathbb{R}$. Let $f_{\text{CNN}}$ have $L$ hidden layers, represented as: for $x \in \mathcal{X}$, $f_{\text{CNN}}(x) := A_{L+1} \circ A_L \circ \cdots \circ A_2 \circ A_1(x)$, where $A_i$'s are either convolutional or downsampling operators. For convolutional layers, $A_i(x) = \sigma(W_i^c x + b_i^c)$, where $W_i^c \in \mathbb{R}^{d_i \times d_{i-1}}$ is a structured sparse Toeplitz matrix from the convolutional filter $\{w_j^{(i)}\}_{j=0}^{s(i)}$ with filter length $s(i) \in \mathbb{N}_+$, $b_i^c \in \mathbb{R}^{d_i}$ is a bias vector, and $\sigma$ is the ReLU activation function. For downsampling layers, $A_i(x) = D_i(x) = (x_{jm_i})_{j=1}^{\lfloor d_{i-1}/m_i \rfloor}$, where $D_i : \mathbb{R}^{d_i \times d_{i-1}}$ is the downsampling operator with scaling parameter $m_i \le d_{i-1}$ in the $i$-th layer. We examine the class of functions represented by CNNs, denoted by $\mathcal{F}_{\text{CNN}}$, defined as:

$$\mathcal{F}_{\text{CNN}} = \{f_{\text{CNN}} \text{ as in defined above with any choice of } A_i, \ i = 1, \ldots, L+1\}.$$

For the specific definition of $\mathcal{F}_{\text{CNN}}$, we refer to [29]'s (4), (5) and (6).

**Assumption 3.5.** Assume that $\mathcal{M}$ and $\mathcal{Z}$ are locally diffeomorphic, that is there exists a map $F : \mathcal{M} \to \mathcal{Z}$ such that at every point $x$ on $\mathcal{M}$, $\det(d F(x)) \ne 0$.

**Theorem 3.6.** *(Approximation Error of Latent Representation). Under Assumption 3.3, 3.4 and 3.5, for $\theta \in (0,1)$, let $d_\theta := \mathcal{O}(d_\mathcal{M} \theta^{-2} \log \frac{d}{\theta})$. For positive integers $M$ and $N$, there exists an encoder $g_{enc}$ and decoder $g_{dec} \in \mathcal{F}_{CNN}(L, S, W)$ s.t.*

$$W_1(g_{enc_\#} Q, P) \le d_\mathcal{M} C (NM)^{-\frac{2(k+1)}{d_\theta}}, \quad W_1(g_{dec_\#} P, Q) \le d_\mathcal{M} C (NM)^{-\frac{2(k+1)}{d_\theta}}.$$

Theorem 3.6 indicates that with an appropriate CNN configuration, the $W_1$ approximation error can be made to reside in a small region, as the best candidate within the function class is indeed capable of approximating the oracle encoder/decoder. In particular, this result indicates that the error magnitude $\varepsilon$ in SDE (5) can be assumed to be small. This allows us to apply the perturbation analysis of the dynamical system defined in Equations (5 - 8) in the following sections.

## 3.2 Latent Representation Errors as Implicit Regularization towards Generalization

In this section, we investigate the impact of latent representation errors on generalization, for the two cases with *zero drift* and *non-zero drift*, respectively. We show that under mild conditions, the *zero-drift* errors can function as a natural form of *implicit regularization*, promoting wider landscapes for improved robustness. Nevertheless, we caution that when latent representation errors have non-zero drift, it could lead to poor regularization with *unstable bias* and degrade world model's generalization, calling for explicit regularization.

To simplify the notation here, we consider the system equations, specifically Equations (5), (6) - (8), as one stochastic system. Let $x_t = (z_t, h_t, \tilde{z}_t, \tilde{s}_t)$ and $B_t = (B_t^{\text{enc}}, B_t^{\text{seq}}, B_t^{\text{pred}}, B_t^{\text{dec}})$:

$$d\, x_t = (g(x_t, t) + \varepsilon\, \sigma(x_t, t))\, dt + \sum_i \bar{g}_i(x_t, t) + \varepsilon\, \bar{\sigma}_i(x_t, t)\, dB_t^i, \tag{9}$$

where $g$, and $\bar{g}_i$ are structured accordingly for the respective components, employing the Einstein summation convention for concise representation. For abuse of notation, $\sigma = (\sigma, 0, 0, 0), \bar{\sigma} = (\bar{\sigma}, 0, 0, 0)$. For a given error magnitude $\varepsilon$, we denote the solution to SDE (9) as $x_t^\varepsilon$. Intuitively, $x_t^\varepsilon$ is the perturbed trajectory of the latent dynamics model. In particular, when $\varepsilon = 0$, indicating that the absence of latent representation error in the model, the solution is denoted as $x_t^0$.

### 3.2.1 The Case with Zero-drift Representation Errors

When the drift coefficient $\sigma = 0$, the latent representation errors correspond to a class of well-behaved stochastic processes. The following result translates the induced perturbation on the stochastic latent

dynamics model's loss function $\mathcal{L}$ to a form of explicit regularization. We assume that $\mathcal{L} \in \mathcal{C}^2$ and depends on $z_t, h_t, \tilde{z}_t, \tilde{s}_t$. Loss functions used in practical implementation, e.g. in DreamerV3, reconstruction loss $J_O$, reward loss $J_R$, consistency loss $J_D$, all satisfy this condition.

**Theorem 3.7.** *(Explicit Effect Induced by Zero-Drift Representation Error) Under Assumptions 3.1 and 3.2 and considering a loss function $\mathcal{L} \in \mathcal{C}^2$, the explicit effects of the zero-drift error can be marginalized out as follows: as $\varepsilon \to 0$,*

$$\mathbb{E}\,\mathcal{L}\left(x_t^{\varepsilon}\right) = \mathbb{E}\,\mathcal{L}(x_t^0) + \mathcal{R} + \mathcal{O}(\varepsilon^3), \tag{10}$$

*where the regularization term $\mathcal{R}$ is given by $\mathcal{R} := \varepsilon\,\mathcal{P} + \varepsilon^2\left(\mathcal{Q} + \frac{1}{2}\,\mathcal{S}\right)$, with*

$$\mathcal{P} := \mathbb{E}\,\nabla\mathcal{L}(x_t^0)^{\top}\,\Phi_t \sum_k \xi_t^k, \tag{11}$$

$$\mathcal{S} := \mathbb{E} \sum_{k_1, k_2} (\Phi_t \xi_t^{k_1})^i \nabla^2 \mathcal{L}(x_t^0, t)\,(\Phi_t \xi_t^{k_2})^j, \tag{12}$$

$$\mathcal{Q} := \mathbb{E}\,\nabla\mathcal{L}(x_t^0)^{\top}\,\Phi_t \int_0^t \Phi_s^{-1}\,\mathcal{H}^k(x_s^0, s)dB_t^k. \tag{13}$$

*Square matrix $\Phi_t$ is the stochastic fundamental matrix of the corresponding homogeneous equation:*

$$d\Phi_t = \frac{\partial \bar{g}_k}{\partial x}(x_t^0, t)\,\Phi_t\,dB_t^k, \quad \Phi(0) = I,$$

*and $\xi_t^k$ is the shorthand for $\int_0^t \Phi_s^{-1} \bar{\sigma}_k(x_s^0, s)dB_t^k$. Additionally, $\mathcal{H}^k(x_s^0, s)$ is represented by for $\sum_{k_1,k_2} \frac{\partial^2 \bar{g}_k}{\partial x^i \partial x^j}(x_s^0, s)\left(\xi_s^{k_1}\right)^i \left(\xi_s^{k_2}\right)^j$.*

The proof is relegated to Appendix B in the Supplementary Materials.

When the loss $\mathcal{L}$ is convex, then its Hessian, $\nabla^2 \mathcal{L}$, is positive semi-definite, which ensures that the term $\mathcal{S}$ is non-negative. *The presence of this Hessian-dependent term $\mathcal{S}$, under latent representation error, implies a tendency towards wider minima in the loss landscape.* Empirical results from [20] indicates that wider minima correlate with improved robustness of implicit regularization during training. This observation also aligns with the theoretical insights in [27] that the introduction of Brownian motion, which is indeed zero-drift by definition, in training RNN models promotes robustness. We note that in addition, when the error $\bar{\sigma}_t(\cdot)$ is too small, the effect of term $\mathcal{S}$ as implicit regularization would not be as significant as desired. Intuitively, this insight resonates with the empirical results in Table 1 that model's robustness gain is not significant when the error induced by small batch sizes is too small.

We remark that the exact loss form treated here is simplified compared to that in the practical implementation of world models, which frequently depends on the probability density functions (PDFs) of $z_t, h_t, \tilde{z}_t, \tilde{s}_t$. In principle, the PDE formulation corresponding to the PDFs of the perturbed $x_t^{\varepsilon}$ can be derived from the Kolmogorov equation of the SDE (9), and the technicality is more involved but can offer more direct insight. We will study this in future work.

### 3.2.2 The Case with Non-Zero-Drift Representation Errors

In practice, latent representation errors may not always exhibit *zero drift* as in idealized noise-injection schemes for deep learning ([27], [2]). When the drift coefficient $\sigma$ is non-zero or a function of input data $h_t$ and $s_t$ in general, the explicit regularization terms induced by the latent representation error may lead to unstable bias in addition to the regularization term $\mathcal{R}$ in Theorem 3.7. With a slight abuse of notation, we denote $\bar{g}_0$ as $g$ from Equation (9) for convenience.

**Corollary 3.8.** *(Additional Bias Induced by Non-Zero Drift Representation Error)*
*Under Assumptions 3.1 and 3.2 and considering a loss function $\mathcal{L} \in \mathcal{C}^2$, the explicit effects of the general form error can be marginalized out as follows as $\varepsilon \to 0$:*

$$\mathbb{E}\,\mathcal{L}\left(x_t^{\varepsilon}\right) = \mathbb{E}\,\mathcal{L}(x_t^0) + \mathcal{R} + \tilde{\mathcal{R}} + \mathcal{O}(\varepsilon^3), \tag{14}$$

*where the additional bias term $\tilde{\mathcal{R}}$ is given by $\tilde{\mathcal{R}} := \varepsilon\,\tilde{\mathcal{P}} + \varepsilon^2\left(\tilde{\mathcal{Q}} + \tilde{\mathcal{S}}\right)$, with*

$$\tilde{\mathcal{P}} := \mathbb{E}\,\nabla\mathcal{L}(x_t^0)^{\top}\,\Phi_t\,\tilde{\xi}_t, \tag{15}$$

$$\tilde{\mathcal{Q}} := \mathbb{E}\,\nabla\mathcal{L}(x_t^0)^{\top}\,\Phi_t \int_0^t \Phi_s^{-1}\,\mathcal{H}^0(x_s^0, s)\,dt, \tag{16}$$

$$\tilde{\mathcal{S}} := \mathbb{E} \sum_k (\Phi_t \tilde{\xi}_t)^i \nabla^2 \mathcal{L}(x_t^0, t)\,(\Phi_t \xi_t^k)^j, \tag{17}$$

258 *and $\tilde{\xi}_t$ being the shorthand for $\int_0^t \Phi_s^{-1} \sigma_k(x_s^0, s) dt$.*

259 The presence of the new bias term $\tilde{\mathcal{R}}$ implies that regularization effects of latent representation error
260 could be unstable. The presence of $\tilde{\xi}$ in $\tilde{\mathcal{P}}$, $\tilde{\mathcal{Q}}$ and $\tilde{\mathcal{S}}$ induces a bias to the loss function with its
261 magnitude dependent on the error level $\varepsilon$, since $\tilde{\xi}$ is a non-zero term influenced on the drift term
262 $\sigma$. This contrasts with the scenarios described in [27] and [2], where the noise injected for implicit
263 regularization follows a zero-mean Gaussian distribution. To modulate the regularization and bias
264 terms $\mathcal{R}$ and $\tilde{\mathcal{R}}$ respectively, we note that a common factor, the fundamental matrix $\Phi$, can be bounded
265 by

$$\mathbb{E} \sup_t \|\Phi_t\|_F^2 \leq \sum_k C \exp \left( C \, \mathbb{E} \sup_t \left\| \frac{\partial g_k}{\partial x}(x_t^0, t) \right\|_F^2 \right) \tag{18}$$

266 which can be shown by using the Burkholder-Davis-Gundy Inequality and Gronwall's Lemma.
267 Based on this observation, we next propose a regularizer on input-output Jacobian norm $\|\frac{\partial g_k}{\partial x}\|_F$ that
268 could modulate the new bias term $\tilde{\mathcal{R}}$ for stabilized implicit regularization.

## 4 Enhancing Predictive Rollouts via Jacobian Regularization

270 In this section, we study the effects of latent representation errors on predictive rollouts using latent
271 state transitions, which happen in the inference phase in world models. We then propose to use
272 Jacobian regularization to enhance the quality of rollouts. In particular, we first obtain an upper bound
273 of state trajectory divergence in the rollout due to the representation error. We show that the error
274 effects on task policy's $Q$ function can be controlled through model's input-output Jacobian norm.

275 In world model learning, the task policy is optimized over the rollouts of dynamics model with the
276 initial latent state $z_0$. Recall that latent representation error is introduced to $z_0$ when latent encoder
277 encodes the initial state $s_0$ from task environment. Intuitively, the latent representation error would
278 propagate under the sequence model and impact the policy learning, which would then affect the
279 generalization capacity through increased exploration.

280 Recall that the sequence model and the transition predictor are given as follows:

$$d\,h_t = f(h_t, \tilde{z}_t, \pi(h_t, \tilde{z}_t))\,dt, \quad d\,\tilde{z}_t = p(h_t)dt + \bar{p}(h_t)\,dB_t, \tag{19}$$

281 with random variables $h_0$, $\tilde{z}_0 + \varepsilon$ as the initial values, respectively. In particular, $\varepsilon$ is a random
282 variable of proper dimension, representing the error from encoder introduced at the initial step. We
283 impose the standard assumption on the error to ensure the well-definedness of the SDEs.

284 Under Assumption 3.1, there exists a unique solution to the SDEs (for Equations 19 with square-
285 integrable $\varepsilon$), denoted as $(h_t^\varepsilon, z_t^\varepsilon)$. In the case of no error introduced, i.e., $\varepsilon = 0$, we denote the
286 solution of the SDEs as $(h_t^0, z_t^0)$ understood as the rollout under the absence of latent representation
287 error. To understand how to modulate impacts of the error in rollouts, our following result gives an
288 upper bound on the expected divergence between the perturbed rollout trajectory $(h_t^\varepsilon, z_t^\varepsilon)$ and the
289 original $(h_t^0, z_t^0)$ over the interval $[0, T]$.

290 **Theorem 4.1.** *(**Bounding trajectory divergence**) For a square-integrable random variable $\varepsilon$, let*
291 $\delta := \mathbb{E} \|\varepsilon\|$ *and* $d_\varepsilon := \mathbb{E} \sup_{t \in [0,T]} \|h_t^\varepsilon - h_t^0\|^2 + \|\tilde{z}_t^\varepsilon - \tilde{z}_t^0\|^2$. *As $\delta \to 0$,*

$$d_\varepsilon \leq \delta\,C\,(\mathcal{J}_0 + \mathcal{J}_1) + \delta^2\,C \exp\left(\mathcal{H}_0\,(\mathcal{J}_0 + \mathcal{J}_1)\right) + \delta^2\,C \exp\left(\mathcal{H}_1\,(\mathcal{J}_0 + \mathcal{J}_1)\right) + \mathcal{O}(\delta^3),$$

292 *where $C$ is a constant dependent on T. $\mathcal{J}_1$ and $\mathcal{J}_2$ are Jacobian-related terms, and $\mathcal{H}_1$ and $\mathcal{H}_2$ are Hessian-*
293 *related terms.*

294 The Jacobian-related terms $\mathcal{J}_1$ and $\mathcal{J}_2$ are defined as $\mathcal{J}_0 := \exp\left(\mathcal{F}_h + \mathcal{F}_z + \mathcal{P}_h\right)$, $\mathcal{J}_1 := \exp\left(\bar{\mathcal{P}}_h\right)$;
295 the Hessian-related terms $\mathcal{H}_0$ and $\mathcal{H}_1$ are defined as $\mathcal{H}_0 := \mathcal{F}_{hh} + \mathcal{F}_{hz} + \mathcal{F}_{zh} + \mathcal{F}_{zz} + \mathcal{P}_{hh}$, $\mathcal{H}_1 := \bar{\mathcal{P}}_{hh}$,
296 where $\mathcal{F}_h$, $\mathcal{F}_z$ are the expected sup Frobenius norm of Jacobians of $f$ w.r.t $h$, $z$, respectively, and
297 $\mathcal{F}_{hh}, \mathcal{F}_{hz}, \mathcal{F}_{zh}, \mathcal{F}_{zz}$ are the corresponding expected sup Frobenius norm of second-order derivatives.
298 Other terms are similarly defined. A detailed description of all terms, can be found in Appendix C.1.

299 Theorem 4.1 correlates with the empirical findings in [14] regarding the diminished predictive
300 accuracy of latent states $\tilde{z}_t$ over the extended horizons. In particular, Theorem 4.1 suggests that the
301 expected divergence from error accumulation hinges on the expected error magnitude, the Jacobian
302 norms within the latent dynamics model and the horizon length $T$.

Our next result reveals how initial latent representation error influences the value function $Q$ during the prediction rollouts, which again verifies that the perturbation is dependent on expected error magnitude, the model's Jacobian norms and the horizon length $T$:

**Corollary 4.2.** *For a square-integrable $\varepsilon$, let $x_t := (h_t, z_t)$. Then, for any action $a \in \mathcal{A}$, the following holds for value function $Q$ almost surely:*

$$Q(x_t^\varepsilon, a) = Q(x_t^0, a) + \frac{\partial}{\partial x} Q(x_t^0, a) \left( \varepsilon^i \partial_i x_t^0 + \frac{1}{2} \varepsilon^i \varepsilon^j \partial_{ij}^2 x_t^0 \right)$$
$$+ \frac{1}{2} (\varepsilon^i \partial_i x_t^0)^\top \frac{\partial^2}{\partial x^2} Q(x_t^0, a) (\varepsilon^i \partial_i x_t^0) + \mathcal{O}(\delta^3),$$

*as $\delta \to 0$, where stochastic processes $\partial_i x_t^0$, $\partial_{ij}^2 x_t^0$ are the first and second derivatives of $x_t^0$ w.r.t $\varepsilon$ and are bounded as follows:*

$$\mathbb{E} \sup_{t \in [0,T]} \left\| \partial_i x_t^0 \right\| \leq C \left( \mathcal{J}_0 + \mathcal{J}_1 \right), \ \mathbb{E} \sup_{t \in [0,T]} \left\| \partial_{ij}^2 x_t^0 \right\| \leq C \exp\left( \mathcal{H}_0 \left( \mathcal{J}_0 + \mathcal{J}_1 \right) \right) + C \exp\left( \mathcal{H}_1 \left( \mathcal{J}_0 + \mathcal{J}_1 \right) \right).$$

This corollary reveals that latent representation errors implicitly encourage exploration of unseen states by inducing a stochastic perturbation in the value function, which again can be regularized through a controlled Jacobian norm.

**Jacobian Regularization against Non-Zero Drift.** The above theoretical results have established a close connection of input-output Jacobian matrices with the stabilized generalization capacity of world models (shown in 18 under non-zero drift form), and perturbation magnitude in predictive rollouts (indicated in the presence of Jacobian terms in Theorem 4.1 and Corollary 4.2.) Based on this, we propose a regularizer on input-output Jacobian norm $\| \frac{\partial g_k}{\partial x} \|_F$ that could modulate $\tilde{\xi}$ ( and in addition $\xi_k$) for stabilized implicit regularization.

The regularized loss function for LDM is defined as follows:

$$\bar{\mathcal{L}}_{\text{dyn}} = \mathcal{L}_{\text{dyn}} + \lambda \| J_\theta \|_F, \tag{20}$$

where $\mathcal{L}_{\text{dyn}}$ is the original loss function for dynamics model, $J_\theta$ denotes the data-dependent Jacobian matrix associated with the $\theta$-parameterized dynamics model, and $\lambda$ is the regularization weight. Our empirical results in 5 with an emphasis on sequential case align with the experimental findings from [18] that Jacobian regularization can enhance robustness against random and adversarial input perturbation in machine learning models.

# 5 Experimental Studies

In this section, experiments are carried out over a number of tasks in Mujoco environments. Due to space limitation, implementation details and additional results, including the standard deviation of the trials, are relegated to Section D in the Appendix.

**Enhanced generalization to unseen noisy states.** We investigated the effectiveness of Jacobian regularization in model trained against a vanilla model during the inference phase with perturbed state images. We consider three types of perturbations: (1) Gaussian noise across the full image, denoted as $\mathcal{N}(\mu_1, \sigma_1^2)$ ; (2) rotation; and (3) noise applied to a percentage of the image, $\mathcal{N}(\mu_2, \sigma_2^2)$. (In Walker task, $\mu_1 = \mu_2 = 0.5, \sigma_2^2 = 0.15$; in Quadruped task, $\mu_1 = 0, \mu_2 = 0.05, \sigma_2^2 = 0.2$.) In each case of perturbations, we examine a collection of noise levels: (1) variance $\sigma^2$ from 0.05 to 0.55; (2) rotation degree $\alpha$ 20 and 30; and (3) masked image percentage $\beta\%$ from 25 to 75.

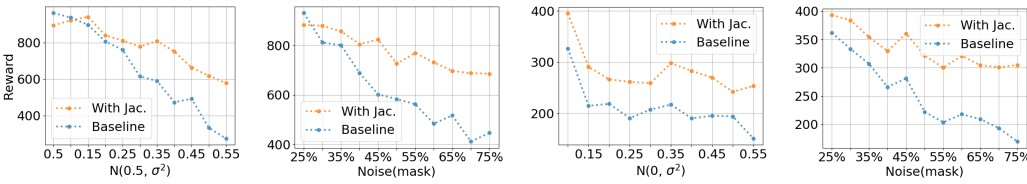

Figure 1: Generalization against increasing degree of perturbation.

It can be seen from Table 3 and Figure 1 that thanks to the adoption of Jacobian regularization in training, the rewards (averaged over 5 trials) are higher compared to the baseline, indicating improved generalization to unseen image states in all cases. The experimental results corroborate the findings in Corollary 3.8 that the regularized Jacobian norm could stabilize the induced implicit regularization.

| | clean | full, $\mathcal{N}(\mu_1, \sigma_1^2)$ | | rotation, $+\alpha°$ | | mask $\beta\%, \mathcal{N}(\mu_2, \sigma_2^2)$ | |
| --- | --- | --- | --- | --- | --- | --- | --- |
| | | $\sigma_1^2 = 0.35$ | $\sigma_1^2 = 0.5$ | $\alpha = 20$ | $\alpha = 30$ | $\beta = 50$ | $\beta = 75$ |
| With Jacobian (Walker) | **967.12** | **742.32** | **618.98** | **423.81** | **226.04** | **725.81** | **685.49** |
| Baseline (Walker) | 966.53 | 615.79 | 333.47 | 391.65 | 197.53 | 583.41 | 446.74 |
| With Jacobian (Quad) | **971.98** | **269.78** | **242.15** | **787.63** | **610.53** | **321.55** | **304.92** |
| Baseline (Quad) | 967.91 | 207.33 | 194.08 | 681.03 | 389.41 | 222.22 | 169.58 |

Table 2: Evaluation on unseen states by various perturbation (Clean means without perturbation). $\lambda = 0.01$.

**Robustness against encoder errors.** Next, we focus on the effects of Jacobian regularization on controlling the error process to the latent states $z$ during training. Since it is very challenging, if not impossible, to characterize the latent representation errors and hence the drift therein explicitly, we consider to evaluate the robustness against two exogenous error signals, namely (1) zero-drift error with $\mu_t = 0, \sigma_t^2$ ($\sigma_t^2 = 5$ in Walker, $\sigma_t^2 = 0.1$ in Quadruped), and (2) non-zero-drift error with $\mu_t \sim [0, 5], \sigma_t^2 \sim [0, 5]$ uniformly. Table 3 shows that the model with regularization can consistently learn policies with high returns and also converges faster, compared to the vanilla case. This corroborates our theoretical findings in Corollary 3.8 that the impacts of error to loss $\mathcal{L}$ can be controlled through the model's Jacobian norm.

| | Zero drift, Walker | | Non-zero drift, Walker | | Zero drift, Quad | | Non-zero drift, Quad | |
| --- | --- | --- | --- | --- | --- | --- | --- | --- |
| | 300k | 600k | 300k | 600k | 600k | 1.2M | 1M | 2M |
| With Jacobian | **666.2** | **966** | **905.7** | **912.4** | **439.8** | **889** | **348.3** | **958.7** |
| Baseline | 24.5 | 43.1 | 404.6 | 495 | 293.6 | 475.9 | 48.98 | 32.87 |

Table 3: Accumulated rewards under additional encoder errors. $\lambda = 0.01$.

**Faster convergence on tasks with extended horizon.** We further evaluate the efficacy of Jacobian regularization in tasks with extended horizon, particularly by extending the horizon length in MuJoCo Walker from 50 to 100 steps. Table 4 shows that the model with regularization converges significantly faster ($\sim 100$K steps) than the case without Jacobian regularization in training. This corroborates results in Theorem 4.1 that regularizing the Jacobian norm can reduce error propagation.

| | Walker 100 len (increased from original 50 len) | | |
| --- | --- | --- | --- |
| Num steps | 100k | 200k | 280k |
| With Jacobian ($\lambda = 0.05$) | **639.1** | **936.3** | 911.1 |
| With Jacobian ($\lambda = 0.1$) | 537.5 | 762.6 | **927.7** |
| Baseline | 582.3 | 571.2 | 886.6 |

Table 4: Accumulated rewards of Walker with extended horizon.

# 6 Conclusion

In this study, we investigate the impacts of latent representation errors on the generalization capacity of world models. We utilize a stochastic differential equation formulation to characterize the effects of latent representation errors as implicit regularization, for both cases with zero-drift errors and with non-zero drift errors. We develop a Jacobian regularization scheme to address the compounding effects of non-zero drift, thereby enhancing training stability and generalization. Our empirical findings validate that Jacobian regularization improves the generalization performance, expanding the applicability of world models in complex, real-world scenarios. Future research is needed to investigate how stabilizing latent errors can enhance generalization across more sophisticated tasks for general non-zero drift cases.

The broader social impact of our work resides in its potential to enhance the robustness and reliability of RL agents deployed in real-world applications. By improving the generalization capacities of world models, our work could contribute to the development of RL agents that perform consistently across diverse and unseen environments. This is particularly relevant in safety-critical domains such as autonomous driving, where reliable agents can provide intelligent and trustworthy decision-making.

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

# Supplementary Materials

In this appendix, we provide the supplementary materials supporting the findings of the main paper on the latent representation of latent representations in world models. The organization is as follows:

- In Section A, we provide proof on showing the approximation capacity of CNN encoder-decoder architecture in latent representation of world models.

- In Section B, we provide proof on implicit regularization of zero-drift errors and additional effects of non-zero-drift errors by showing a proposition on the general form.

- In Section C, we provide proof on showing the effects of non-zero-drift errors during predictive rollouts by again showing a result on the general form.

- In Section D, we provide additional results and implementation details on our empirical studies.

# A Approximation Power of Latent Representation with CNN Encoder and Decoder

To mathematically describe this *intrinsic lower-dimensional geometric structure*, for an integer $k > 0$ and $\alpha \in (0, 1]$, we consider the notion of smooth manifold (in the $\mathcal{C}^{k,\alpha}$ sense), formally defined by

**Definition A.1** ($\mathcal{C}^{k,\alpha}$ manifold). A $\mathcal{C}^{k,\alpha}$ manifold $\mathcal{M}$ of dimension $n$ is a topological manifold (i.e. a topological space that is locally Euclidean, with countable basis, and Hausdorff) that has a $\mathcal{C}^{k,\alpha}$ structure $\Xi$ that is a collection of coordinate charts $\{U_\alpha, \psi_\alpha\}_{\alpha \in A}$ where $U_\alpha$ is an open subset of $\mathcal{M}$, $\psi_\alpha : U_\alpha \to V_\alpha \subseteq \mathbb{R}^n$ such that

- $\bigcup_{\alpha \in A} U_\alpha \supseteq \mathcal{M}$, meaning that the the open subsets form an open cover,

- Each chart $\psi_\alpha$ is a diffeomorphism that is a smooth map with smooth inverse (in the $\mathcal{C}^{k,\alpha}$ sense),

- Any two charts are $\mathcal{C}^{k,\alpha}$-compatible with each other, that is for all $\alpha_1, \alpha_2 \in A$, $\psi_{\alpha_1} \circ \psi_{\alpha_2}^{-1} :$ $\psi_{\alpha_2}(U_{\alpha_1} \cap U_{\alpha_2}) \to \psi_{\alpha_1}(U_{\alpha_1} \cap U_{\alpha_2})$ is $\mathcal{C}^{k,\alpha}$.

Intuitively, a $\mathcal{C}^{k,\alpha}$ manifold is a generalization of Euclidean space by allowing additional spaces with nontrivial global structures through a collection of charts that are diffeomorphisms mapping open subsets from the manifold to open subsets of euclidean space. For technical utility, the defined charts allow to transfer most familiar real analysis tools to the manifold space. For more references, see [25].

**Definition A.2** (Riemannian volume form). Let $\mathcal{X}$ be a smooth, oriented $d$-dimensional manifold with Riemannian metric $g$. A volume form $d\mathrm{vol}_\mathcal{M}$ is the canonical volume form on $\mathcal{X}$ if for any point $x \in \mathcal{X}$, for a chosen local coordinate chart $(x_1, ..., x_d)$, $d\mathrm{vol}_\mathcal{M} = \sqrt{\det g_{ij}}\, dx_1 \wedge ... \wedge dx_d$, where $g_{ij}(x) := g\left(\frac{\partial}{\partial x_i}, \frac{\partial}{\partial x_j}\right)(x)$.

Then the induced volume measure by the canonical volume form $d\mathrm{vol}_\mathcal{X}$ is denoted as $\mu_\mathcal{X}$, defined by $\mu_\mathcal{X} : A \mapsto \int_A d\mathrm{vol}_\mathcal{X}$, for any Borel-measurable subset $A$ on the space $\mathcal{X}$. For more references, see [9].

We recall the latent representation problem defined in the main paper.

Consider the state space $\mathcal{S} \subset \mathbb{R}^{d_\mathcal{S}}$ and the latent space $\mathcal{Z}$. Consider a state probability measure $Q$ on the state space $\mathcal{S}$ and a probability measure $P$ on the latent space $\mathcal{Z}$.

**Assumption A.3.** (Latent manifold assumption) For a positive integer $k$, there exists a $d_\mathcal{M}$-dimensional $\mathcal{C}^{k,\alpha}$ submanifold $\mathcal{M}$ (with $\mathcal{C}^{k+3,\alpha}$ boundary) with Riemannian metric $g$ and has positive reach and also isometrically embedded in the state space $\mathcal{S} \subset \mathbb{R}^{d_\mathcal{S}}$ and $d_\mathcal{M} \ll d_\mathcal{S}$, where the state probability measure is supported on. In addition, $\mathcal{M}$ is a compact, orientable, connected manifold.

**Assumption A.4.** (Smoothness of state probability measure) $Q$ is a probability measure supported on $\mathcal{M}$ with its Radon-Nikodym derivative $q \in \mathcal{C}^{k,\alpha}(\mathcal{M}, \mathbb{R})$ w.r.t $\mu_\mathcal{M}$.

Let $\mathcal{Z}$ be a closed ball in $\mathbb{R}^{d_\mathcal{M}}$, that is $\{x \in \mathbb{R}^{d_\mathcal{M}} : \|x\| \leq 1\}$. $P$ is a probability measure supported on $\mathcal{Z}$ with its Radon-Nikodym derivative $p \in \mathcal{C}^{k,\alpha}(\mathcal{Z}, \mathbb{R})$ w.r.t $\mu_\mathcal{Z}$.

We consider a general CNN function $f_{\mathrm{CNN}} : \mathcal{X} \to \mathbb{R}$. Let $f_{\mathrm{CNN}}$ have $L$ hidden layers, represented as:

$$f_{\mathrm{CNN}}(x) = A_{L+1} \circ A_L \circ \cdots \circ A_2 \circ A_1(x), \quad x \in \mathcal{X},$$

where $A_i$'s are either convolutional or downsampling operators. For convolutional layers,

$$A_i(x) = \sigma(W_i^c x + b_i^c),$$

where $W_i^c \in \mathbb{R}^{d_i \times d_{i-1}}$ is a structured sparse Toeplitz matrix from the convolutional filter $\{w_j^{(i)}\}_{j=0}^{s(i)}$ with filter length $s(i) \in \mathbb{N}_+$, $b_i^c \in \mathbb{R}^{d_i}$ is a bias vector, and $\sigma$ is the ReLU activation function.

For downsampling layers,

$$A_i(x) = D_i(x) = (x_{jm_i})_{j=1}^{\lfloor d_{i-1}/m_i \rfloor},$$

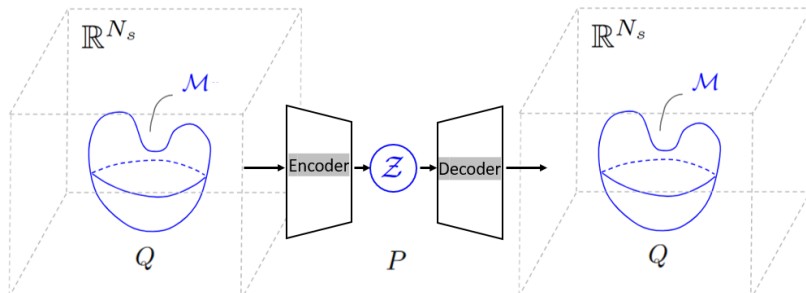

Figure 2: *Latent Representation Problem*: The left and right denote the manifold $\mathcal{M}$ with lower dim $d_{\mathcal{M}}$ embedded in a larger Euclidean space, with latent space $Z$ a $d_{\mathcal{M}}$-dimensional ball in middle. Encoder and decoder as maps respectively pushing forward Q to P and P to Q.

where $D_i : \mathbb{R}^{d_i \times d_{i-1}}$ is the downsampling operator with scaling parameter $m_i \leq d_{i-1}$ in the $i$-th layer. The convolutional and downsampling operations are elaborated in Appendix [63]. We examine the class of functions represented by CNNs, denoted by $\mathcal{F}_{\text{CNN}}$, defined as:

$$\mathcal{F}_{\text{CNN}} = \{f_{\text{CNN}} \text{ as in defined above with any choice of } A_i,\ i = 1, \ldots, L+1\}.$$

For more details in the definitions of CNN functions, we refer to [29].

**Assumption A.5.** Assume that $\mathcal{M}$ and $\mathcal{Z}$ are locally diffeomorphic, that is there exists a map $F : \mathcal{M} \to \mathcal{Z}$ such that at every point $x$ on $\mathcal{M}$, $\det(d\,F(x)) \neq 0$.

**Theorem A.6.** *(Approximation Error of Latent Representation). Under Assumption A.3, A.4 and A.5, for $\theta \in (0,1)$, let $d_\theta = \mathcal{O}(d_{\mathcal{M}}\theta^{-2} \log \frac{d}{\theta})$. For positive integers $M$ and $N$, there exists an encoder $g_{enc}$ and decoder $g_{dec} \in \mathcal{F}_{CNN}(L, S, W)$ s.t.*

$$W_1(g_{enc_\#}Q, P) \leq d_{\mathcal{M}}C(NM)^{-\frac{2(k+1)}{d_\theta}},$$
$$W_1(g_{dec_\#}P, Q) \leq d_{\mathcal{M}}C(NM)^{-\frac{2(k+1)}{d_\theta}}.$$

The primary challenge to show Theorem A.6 is in demonstrating the existence of oracle encoder and decoder maps. These maps, denoted as $g^*_{\text{enc}} : \mathcal{M} \to \mathcal{Z}$ and $g^*_{\text{dec}} : \mathcal{Z} \to \mathcal{M}$ respectively, must satisfy

$$g^*_{\text{enc}\#}\, Q = P, \quad g^*_{\text{dec}\#}\, P = Q. \tag{21}$$

and importantly they have the proper smoothness guarantee, namely $g^*_{\text{enc}} \in \mathcal{C}^{k+1,\alpha}(\mathcal{M}, \mathcal{Z})$ and $g^*_{\text{dec}} \in \mathcal{C}^{k+1,\alpha}(\mathcal{Z}, \mathcal{M})$. Proposition A.7 shows the existence of such oracle map(s).

**Proposition A.7** ($\mathcal{C}^{k,\alpha}$, compact). *Let $\mathcal{M}, \mathcal{N}$ be compact, oriented $d$-dimensional Riemannian manifolds with $\mathcal{C}^{k+3,\alpha}$ boundary with the volume measure $\mu_{\mathcal{M}}$ and $\mu_{\mathcal{N}}$ respectively. Let $Q, P$ be distributions supported on $\mathcal{M}, \mathcal{N}$ respectively with their $\mathcal{C}^{k,\alpha}$ density functions $q, p$, that is $Q, P$ are probability measures supported on $\mathcal{M}, \mathcal{N}$ with their Radon-Nikodym derivatives $q \in \mathcal{C}^{k,\alpha}(\mathcal{M}, \mathbb{R})$ w.r.t $\mu_{\mathcal{M}}$ and $p \in \mathcal{C}^{k,\alpha}(\mathcal{N}, \mathbb{R})$ w.r.t $\mu_{\mathcal{N}}$. Then, there exists a $\mathcal{C}^{k+1,\alpha}$ map $g : \mathcal{N} \to \mathcal{M}$ such that the pushforward measure $g_\#P = Q$, that is for any measurable subset $A \in \mathcal{B}(\mathcal{M})$, $Q(A) = P(g^{-1}(A))$.*

*Proof.* (*Proposition A.7*) Let $\omega := p\,d\text{vol}_{\mathcal{N}}$, then $\omega$ is a $\mathcal{C}^{k,\alpha}$ volume form on $\mathcal{N}$, as $p \in \mathcal{C}^{k,\alpha}$ and for any point $x \in \mathcal{N}$, we have $p(x) > 0$. In addition, $\int_{\mathcal{N}} \omega = \int_{\mathcal{N}} p\,d\text{vol}_{\mathcal{N}} = \int_{\mathcal{N}} p\,d\mu_{\mathcal{N}} = P(\mathcal{N}) = 1$. Similarly, let $\eta := q\,d\text{vol}_{\mathcal{M}}$ a $\mathcal{C}^{k,\alpha}$ volume form on $\mathcal{M}$ and $\int_{\mathcal{M}} \eta = 1$.

Let $F : \mathcal{N} \to \mathcal{M}$ be an orientation-preserving local diffeomorphism, we then have $\det(dF) > 0$ everywhere on $\mathcal{N}$.
As $\mathcal{N}$ is compact and $\mathcal{M}$ is connected by assumption, $F$ is a covering map, that is for every point $x \in \mathcal{M}$, there exists an open neighborhood $U_x$ of $x$ and a discrete set $D_x$ such that $F^{-1}(U) = \sqcup_{\alpha \in D} V_\alpha \subset \mathcal{N}$ and $F|_{V_\alpha} = V_\alpha \to U$ is a diffeomorphism. Furthermore, $|D_x| = |D_y|$ for any points $x, y \in \mathcal{M}$. In addition, $|D_x|$ is finite from the compactness of $\mathcal{N}$.

525 Let $\bar{\eta}$ be the pushforward of $\omega$ via $F$, defined by for any point $x \in \mathcal{M}$ and a neighborhood $U_x$,

$$\bar{\eta}(x) := \frac{1}{|D_x|} \sum_{\alpha \in D_x} \left( F\big|_{V_\alpha}^{-1} \right)^* \omega\big|_{V_\alpha}. \tag{22}$$

526 $\bar{\eta}$ is well-defined as it is not dependent on the choice of neighborhoods and the sum and $\frac{1}{|D_x|}$ are

527 always finite. Furthermore, $\bar{\eta}$ is a $\mathcal{C}^{k,\alpha}$ volume form on $\mathcal{M}$, as $p \circ \left( F\big|_{V_\alpha}^{-1} \right)$ is $\mathcal{C}^{k,\alpha}$.

528

529 Notice that $F\big|_{V_\alpha}^{-1}$ is orientation-preserving as $\det dF\big|_{V_\alpha}^{-1} = \frac{1}{\det dF\big|_{V_\alpha}} > 0$ everywhere on $V_\alpha$.

530 In addition, $F\big|_{V_\alpha}^{-1}$ is proper: as for any compact subset $K$ of $\mathcal{N}$, $K$ is closed; and as $F\big|_{V_\alpha}^{-1}$

531 is continuous, the preimage of $K$ via $F\big|_{V_\alpha}^{-1}$ a closed subset of $\mathcal{M}$ which is compact, then the

532 preimage of $K$ must also be compact. Hence, $F\big|_{V_\alpha}^{-1}$ is proper. As every $F\big|_{V_\alpha}^{-1}$ is proper,

533 orientation-preserving and surjective, then $c := \deg(F\big|_{V_\alpha}^{-1}) = 1$.

534 Then, $\int_{\mathcal{M}} \bar{\eta} = c \int_{\mathcal{N}} \omega = 1$.

535

536 As we have shown that $\eta$ and $\bar{\eta} \in \mathcal{C}^{k,\alpha}$ and $\int_{\mathcal{M}} \bar{\eta} = \int_{\mathcal{M}} \eta$, by [6], there exists a diffeomorphism

537 $\psi : \mathcal{M} \to \mathcal{M}$ fixing on the boundary such that $\psi^* \eta = \bar{\eta}$, where $\psi, \psi^{-1} \in \mathcal{C}^{k+1,\alpha}$.

538 Let $g := \psi \circ F$, then it holds that $g^* \eta = (\psi \circ F)^* \eta = F^* \circ \psi^* \eta = F^* \bar{\eta} = \omega$.

539 Then, for any measurable subset $A$ on the manifold $\mathcal{M}$, we verify that $Q(A) = \int_A \eta =$

540 $\int_{g^{-1}(A)} g^* \eta = \int_{g^{-1}(A)} \omega = \int_{g^{-1}(A)} p \, d\mathrm{vol}_{\mathcal{N}} = \int_{g^{-1}(A)} p \, d\mu_{\mathcal{N}} = P(g^{-1}(A))$.

541

542 Hence, we have shown the existence by an explicit construction. As $\psi \in \mathcal{C}^{k+1,\alpha}$, and $F \in \mathcal{C}^\infty$, then

543 we have $g \in \mathcal{C}^{k+1,\alpha}$. $\qquad\square$

544 We are now ready to show Theorem A.6 with the existence of oracle map and the low-dimensional

545 approximation results from [29].

546 *Proof. (Theorem A.6)* For encoder, from Proposition A.7, there exists an $\mathcal{C}^{k+1,\alpha}$ oracle map $g :$

547 $\mathcal{M} \to \mathcal{Z}$ such that the pushforward measure $g_\# Q = P$. Then,

$$W_1((g_{\mathrm{enc}})_\# Q \,,\, P) = W_1((g_{\mathrm{enc}})_\# Q \,,\, g_\# Q)$$
$$= \sup_{f \in \mathrm{Lip}_1(\mathcal{Z})} \left| \int_{\mathcal{Z}} f(y) \, d((g_{\mathrm{enc}})_\# Q) - \int_{\mathcal{Z}} f(y) \, d(g_\# Q) \right|$$
$$\leq \sup_{f \in \mathrm{Lip}_1(\mathcal{Z})} \int_{\mathcal{M}} |f \circ g_{\mathrm{enc}}(x) - f \circ g(x)| \, dQ$$
$$\leq \int_{\mathcal{M}} \|g_{\mathrm{enc}}(x) - g(x)\| \, dQ$$
$$\leq d_{\mathcal{M}} C(NM)^{-\frac{2(k+1)}{d_\theta}},$$

548 where the last inequality follows from the special case $\rho = 0$ of Theorem 2.4 in [29].

549 Similarly, for decoder, from Proposition A.7, there exists an $\mathcal{C}^{k+1,\alpha}$ oracle map $\bar{g} : \mathcal{Z} \to \mathcal{M}$ such

550 that the pushforward measure $\bar{g}_\# P = Q$.

$$W_1((g_{\mathrm{dec}})_\# P \,,\, Q) = W_1((g_{\mathrm{dec}})_\# P \,,\, \bar{g}_\# P)$$
$$\leq \int_{\mathcal{Z}} \|g_{\mathrm{dec}}(y) - \bar{g}(y)\| \, dP$$
$$\leq d_{\mathcal{M}} C(NM)^{-\frac{2(k+1)}{d_\theta}}.$$

551 $\qquad\square$

# B  Explicit Regularization of Latent Representation Error in World Model Learning

We recall the SDEs for latent dynamics model defined in the main paper. Consider a complete, filtered probability space $(\Omega, \mathcal{F}, \{\mathcal{F}_t\}_{t \in [0,T]}, \mathbb{P})$ where independent standard Brownian motions $B_t^{\mathrm{enc}}, B_t^{\mathrm{pred}}, B_t^{\mathrm{seq}}, B_t^{\mathrm{dec}}$ are defined such that $\mathcal{F}_t$ is their augmented filtration, and $T \in \mathbb{R}$ as the time length of the task environment. We consider the stochastic dynamics of LDM through the following coupled SDEs after error perturbation:

$$d\, z_t = (q_{\mathrm{enc}}(h_t, s_t) + \sigma(h_t, s_t))\, dt + (\bar{q}_{\mathrm{enc}}(h_t, s_t) + \bar{\sigma}(h_t, s_t))\, dB_t^{\mathrm{enc}}, \tag{23}$$

$$d\, h_t = f(h_t, z_t, \pi(h_t, z_t))\, dt + \bar{f}(h_t, z_t, \pi(h_t, z_t))\, dB_t^{\mathrm{seq}} \tag{24}$$

$$d\, \tilde{z}_t = p(h_t)\, dt + \bar{p}(h_t)\, dB_t^{\mathrm{pred}}, \tag{25}$$

$$d\, \tilde{s}_t = q_{\mathrm{dec}}(h_t, \tilde{z}_t)\, dt + \bar{q}_{\mathrm{dec}}(h_t, \tilde{z}_t)\, dB_t^{\mathrm{dec}}, \tag{26}$$

where $\pi(h, \tilde{z})$ is a policy function as a local maximizer of value function and the stochastic process $s_t$ is $\mathcal{F}_t$-adapted.

As discussed in the main paper, our analysis applies to a common class of world models that uses Gaussian distributions parameterized by neural networks' outputs for $z, \tilde{z}, \tilde{s}$. Their distributions are not non-Gaussian in general.

For example, as $z$ is conditional Gaussian and its mean and variance are random variables which are learned by the encoder from r.v.s $s$ and $h$ as inputs, thus rendering $z$ non-Gaussian. However, $z$ is indeed Gaussian when the inputs are known. Under this conditional Gaussian class of world models, to see that the continuous formulation of latent dynamics model can be interrupted as SDEs, one notices that SDEs with coefficient functions of known inputs are indeed Gaussian, matching to this class of world models. Formally, in the context of $z$ without latent representation error:

**Proposition B.1.** *(Latent states SDE with known inputs is Gaussian)*
*For the latent state process $z_{t \in [0,T]}$ without error,*

$$d\, z_t = q_{enc}(h_t, s_t)\, dt + \bar{q}_{enc}(h_t, s_t))dB_t^{enc}, \tag{27}$$

*with zero initial value. Given known $h_{t \in [0,T]}$ and $s_{t \in [0,T]}$, the process $z_t$ is a Gaussian process. Furthermore, for any $t \in [0,T]$, $z_t$ follows a Gaussian distribution with mean $\mu_t = \int_0^t q_{enc}(h_s, s_s)ds$ and variance $\sigma_t^2 = \int_0^t \bar{q}_{enc}(h_s, s_s)^2 ds$.*

*Proof.* Proof follows from Proposition 7.6 in [30]. $\qquad\qquad\qquad\qquad\qquad\qquad\square$

Next, we recall our assumptions from the main text:

**Assumption B.2.** The drift coefficient functions $q_{\mathrm{enc}}$, $f$, $p$ and $q_{\mathrm{dec}}$ and the diffusion coefficient functions $\bar{q}_{\mathrm{enc}}$, $\bar{p}$ and $\bar{q}_{\mathrm{dec}}$ are bounded and Borel-measurable over the interval $[0,T]$, and of class $\mathcal{C}^3$ with bounded Lipschitz continuous partial derivatives. The initial values $z_0, h_0, \tilde{z}_0, \tilde{s}_0$ are square-integrable random variables.

**Assumption B.3.** $\sigma$ and $\bar{\sigma}$ are bounded and Borel-measurable and are of class $\mathcal{C}^3$ with bounded Lipschitz continuous partial derivatives over the interval $[0,T]$.

One of our main results is the following:

**Theorem B.4.** *(Explicit Regularization Induced by Zero-Drift Representation Error)*
*Under Assumption B.2 and B.3 and considering a loss function $\mathcal{L} \in \mathcal{C}^2$, the explicit effects of the zero-drift error can be marginalized out as follows:*

$$\mathbb{E}\, \mathcal{L}\left(x_t^{\varepsilon}\right) = \mathbb{E}\, \mathcal{L}(x_t^0) + \mathcal{R} + \mathcal{O}(\varepsilon^3), \tag{28}$$

587    *as $\varepsilon \to 0$, where the regularization term $\mathcal{R}$ is given by $\mathcal{R} := \varepsilon \mathcal{P} + \varepsilon^2 \left(\mathcal{Q} + \frac{1}{2}\mathcal{S}\right).$*

588    *Each term of $\mathcal{R}$ is as follows:*

$$\mathcal{P} := \mathbb{E}\, \nabla\mathcal{L}(x_t^0)^\top \Phi_t \sum_k \xi_t^k, \tag{29}$$

$$\mathcal{Q} := \mathbb{E}\, \nabla\mathcal{L}(x_t^0)^\top \Phi_t \int_0^t \Phi_s^{-1}\, \mathcal{H}^k(x_s^0, s) dB_t^k, \tag{30}$$

$$\mathcal{S} := \mathbb{E} \sum_{k_1, k_2} (\Phi_t \xi_t^{k_1})^i \nabla^2 \mathcal{L}(x_t^0, t)\, (\Phi_t \xi_t^{k_2})^j, \tag{31}$$

589    *where square matrix $\Phi_t$ is the stochastic fundamental matrix of the corresponding homogeneous*
590    *equation:*

$$d\Phi_t = \frac{\partial \bar{g}_k}{\partial x}(x_t^0, t)\, \Phi_t\, dB_t^k, \quad \Phi(0) = I,$$

591    *and $\xi_t^k$ is as the shorthand for $\int_0^t \Phi_s^{-1} \bar\sigma_k(x_s^0, s) dB_t^k$. Additionally, $\mathcal{H}^k(x_s^0, s)$ is represented by for*
592    *$\sum_{k_1, k_2} \frac{\partial^2 \bar{g}_k}{\partial x^i \partial x^j}(x_s^0, s) \left(\xi_s^{k_1}\right)^i \left(\xi_s^{k_2}\right)^j.$*

593    Before proving Theorem B.4, we first show Proposition B.5 on the general case of perturbation to the
594    stochastic system. Consider the following perturbed system given by

$$d x_t = (g_0(x_t, t) + \varepsilon\, \eta_0(x_t, t))\, dt + \sum_{k=1}^m (g_k(x_t, t) + \varepsilon\, \eta_k(x_t, t))\, dB_t^k \tag{32}$$

595    with initial values $x(0) = x_0,$

596    **Proposition B.5.** *Suppose that $f$ is a real-valued function that is $\mathcal{C}^2$. Then it holds that, with*
597    *probability 1, as $\varepsilon \to 0$, for $t \in [0, T]$,*

$$f\left(x_t^\varepsilon\right) = f\left(x_t^0\right) + \varepsilon \nabla f\left(x_t^0\right)^\top \partial_\varepsilon x_t^0 + \varepsilon^2 \left(\nabla f\left(x_t^0\right)^\top \partial_\varepsilon^2 x_t^0 + \frac{1}{2}\partial_\varepsilon x_t^{0\top} \nabla^2 f\left(x_t^0\right) \partial_\varepsilon x_t^0\right) + \mathcal{O}\left(\varepsilon^3\right), \tag{33}$$

598    *where the stochastic process $x_t^0$ is the solution to SDE 32 with $\varepsilon = 0$, with its first and second-order*
599    *derivatives w.r.t $\varepsilon$ denoted as $\partial_\varepsilon x_t^0, \partial_\varepsilon^2 x_t^0$.*
600    *Furthermore, it holds that $\partial_\varepsilon x_t^0, \partial_\varepsilon^2 x_t^0$ satisfy the following SDEs with probability 1,*

$$d\,\partial_\varepsilon x_t^0 = \left(\frac{\partial g_k}{\partial x}\left(x_t^0, t\right) \partial_\varepsilon x_t^0 + \eta_k\left(x_t^0, t\right)\right) dB_t^k,$$
$$d\,\partial_\varepsilon^2 x_t = \left(\Psi_k\left(\partial_\varepsilon x_t^0, x_t^0, t\right) + 2\frac{\partial \eta_k}{\partial x}\left(x_t^0, t\right) \partial_\varepsilon x_t^0 + \frac{\partial g_k}{\partial x}\left(x_t^0, t\right) \partial_\varepsilon^2 x_t^0\right) dB_t^k, \tag{34}$$

601    *with initial values $\partial_\varepsilon x(0) = 0, \partial_\varepsilon^2 x(0) = 0$, where*

$$\Psi_k : (\partial_\varepsilon x, x, t) \mapsto \partial_\varepsilon x^i \frac{\partial g_k}{\partial x^i \partial x^j}(x, t)\partial_\varepsilon x^j,$$

602    *for $k = 0, 1, ..., m$.*

603    *Proof.* We first apply the stochastic version of perturbation theory to SDE 32. For brevity, we will
604    write $t$ as $B_t^0$ and use Einstein summation convention. Hence, SDE 32 is rewritten as

$$dx_t = \gamma_k^\varepsilon(x_t, t)\, dB_t^k, \tag{35}$$

605    with initial value $x(0) = x_0$.

606    *Step 1*: We begin with the corresponding systems to derive the SDEs that characterize $\partial_\varepsilon x_t^\varepsilon$ and $\partial_\varepsilon^2 x_t^\varepsilon$.
607    Our main tool is an important result on smoothness of solutions w.r.t. initial data from Theorem 3.1
608    from Section 2 in [17].

609    For $\partial_\varepsilon x$, consider the SDEs

$$d\, x_t = \gamma_k^\varepsilon(x_t, t)\, dB_t^k, \tag{*}$$
$$d\, \varepsilon_t = 0,$$

with initial values $x_{(0)} = x_0, \varepsilon(0) = \varepsilon$. From an application of Theorem 3.1 from Section 2 in [17] on *, we have $\partial_\varepsilon x$ that satisfies the following SDE with probability 1:

$$d\,\partial_\varepsilon x_t = (\alpha_k^\varepsilon(x_t, t)\,\partial_\varepsilon x_t + \eta_k(x_t, t))\,dB_t^k, \tag{36}$$

with initial value $\partial_\varepsilon x_0 = 0 \in \mathbb{R}^n$, with probability 1, where $x_t$ is the solution to Equation (35) and the functions $\alpha_k^\varepsilon$ are given by

$$\alpha_k^\varepsilon : (x, t) \mapsto \frac{\partial g_k}{\partial x^j}(x, t) + \varepsilon \frac{\partial \eta_k}{\partial x^j}(x, t),$$

where $k = 0, ..., m$.

To characterize $\partial_\varepsilon^2 x_t$, consider the following SDEs

$$\begin{aligned}
d\,x_t &= \gamma_k^\varepsilon(x_t, t)\,dB_t^k, \\
d\,\partial_\varepsilon x_t &= (\alpha_k^\varepsilon(x_t, t)\,\partial_\varepsilon x_t + \eta_k(x_t, t))\,dB_t^k, \\
d\,\varepsilon_t &= 0,
\end{aligned} \tag{**}$$

with initial value $x(0) = x_0, \partial_\varepsilon x(0) = 0, \varepsilon(0) = \varepsilon$.

From a similar application of Theorem 3.1 from Section 2 in [17], the second derivative $\partial_\varepsilon^2 x$ satisfies the following SDE with probability 1:

$$d\,\partial_\varepsilon^2 x_t = \left(\beta_k^\varepsilon(\partial_\varepsilon x_t, x_t, t) + 2\frac{\partial \eta_k}{\partial x}(x_t, t)\,\partial_\varepsilon x_t + \alpha_k^\varepsilon(x_t, t)\,\partial_\varepsilon^2 x_t\right)dB_t^k, \tag{37}$$

with initial value $\partial_\varepsilon^2 x(0) = 0 \in \mathbb{R}^n$, where $\partial_\varepsilon x_t$ is the solution to Equation(36), $x(t)$ is the solution to Equation (35), and the functions

$$\beta_k^\varepsilon : (\partial_\varepsilon x, x, t) \mapsto \partial_\varepsilon x^j \left(\frac{\partial g_k^i}{\partial x^l \partial x^j}(x, t) + \varepsilon \frac{\partial \eta_k^i}{\partial x^l \partial x^j}(x, t)\right)\partial_\varepsilon x^l, \text{ where } k = 0, ..., m.$$

When $\varepsilon = 0$ in the obtained SDEs (35), (36) and (37), the corresponding solutions of which are $x_t^0, \partial_\varepsilon x_t^0, \partial_\varepsilon^2 x_t^0$, we now have the following:

$$d\,x_t^0 = g_k(x_t^0, t)\,dB_t^k, \tag{38}$$

$$d\,\partial_\varepsilon x_t^0 = \left(\frac{\partial g_k}{\partial x}(x_t^0, t)\,\partial_\varepsilon x^0 + \eta_k(x_t^0, t)\right)dB_t^k, \tag{39}$$

$$d\,\partial_\varepsilon^2 x_t^0 = \left(\Psi_k(\partial_\varepsilon x_t^0, x_t^0, t) + 2\frac{\partial \eta_k}{\partial x}(x_t^0, t)\,\partial_\varepsilon x_t^0 + \frac{\partial g_k}{\partial x}(x_t^0, t)\,\partial_\varepsilon^2 x_t^0\right)dB_t^k, \tag{40}$$

with initial values $x(0) = x_0, \partial_\varepsilon x(0) = 0, \partial_\varepsilon^2 x(0) = 0$. In particular, $\Psi_k := \beta_k^0$ is given by

$$(\partial_\varepsilon x, x, t) \mapsto \partial_\varepsilon x^i \frac{\partial g_k}{\partial x^i \partial x^i}(x, t)\partial_\varepsilon x^j.$$

*Step 2*: For the next step, we show that the solutions $x_t^0, \partial_s x_t^0, \partial_\varepsilon^2 x_t^0$ are indeed bounded by proving the following lemma B.6:

**Lemma B.6.**
$$\mathbb{E} \sup_{t \in [0,T]} \left\|x_t^0\right\|^2, \ \mathbb{E} \sup_{t \in [0,T]} \left\|\partial_\varepsilon x_t^0\right\|^2, and \ \mathbb{E} \sup_{t \in [0,T]} \left\|\partial_\varepsilon^2 x_t^0\right\|^2 \ are \ bounded.$$

*Proof.* To simplify the notations, we take the liberty to write constants as $C$ and notice that $C$ is not necessarily identical in its each appearance.

(1) We first show that $\mathbb{E} \sup_{t \in [0,T]} \left\|x_t^0\right\|^2$ is bounded.

From Equation (38), we have that

$$x_t^0 = x_0 + \int_0^t g_k(x_\tau, \tau)\,dB_\tau^k.$$

By Jensen's inequality. it holds that

$$\mathbb{E} \sup_{t \in [0,T]} \|x_t\|^2 \leq C \mathbb{E} \|x_0\|^2 + C \mathbb{E} \sup_{t \in [0,T]} \left\| \int_0^t g_k \left( x_\tau^0, \tau \right) dB_\tau^k \right\|^2. \tag{41}$$

For the second term on the right hand side, it is a sum over $k$ from 0 to $m$ by Einstein notation.

For $k = 0$, recall that we write $t$ as $B_t^0$ :

$$\mathbb{E} \sup_{t \in [0,T]} \left\| \int_0^t g_0 \left( x_\tau^0, \tau \right) d\tau \right\|^2 \leq C \mathbb{E} \sup_{t \in [0,T]} t \int_0^t \left\| g_0 \left( x_\tau^0, \tau \right) \right\|^2 d\tau, \tag{i}$$

$$\leq C \mathbb{E} \sup_{t \in [0,T]} \int_0^t C \left( 1 + \left\| x_\tau^0 \right\| \right)^2 d\tau, \tag{ii}$$

$$\leq C + C \int_0^T \mathbb{E} \sup_{s \in [0,\tau]} \left\| x_s^0 \right\|^2 d\tau, \tag{iii}$$

where we used Jensen's inequality, the assumption on the linear growth, the inequality property of sup and Fubini's theorem, respectively.

For $k$ is equal to $1, \ldots, m$,

$$\mathbb{E} \sup_{t \in [0,T]} \left\| \int_0^t g_1 \left( x_{\tau,\tau}^0, \tau \right) dB_\tau \right\|^2 \leq C \mathbb{E} \int_0^T \left\| g_1 \left( x_\tau^0, \tau \right) \right\|^2 d\tau, \tag{iv}$$

$$\leq C + C \int_0^T \mathbb{E} \sup_{s \in [0,\tau]} \left\| x_s^0 \right\| d\tau, \tag{v}$$

where (iv) holds from the Burkholder-Davis-Gundy inequality as $\int_0^t g_k \left( x_\tau^0, \tau \right) dB_\tau$ is a continuous local martingale with respect to the filtration $\mathcal{F}_t$; and then one can obtain (v) by following a similar reasoning of (ii) and (iii).

Hence, now from the previous inequality (41),

$$\mathbb{E} \sup_{t \in [0,T]} \left\| x_t^0 \right\|^2 \leq \mathbb{E} \|x_0\|^2 + C + C \int_0^T \mathbb{E} \sup_{s \in [0,\tau]} \left\| x_s^0 \right\| d\tau.$$

By the Gronwall's lemma, it holds true that

$$\mathbb{E} \sup_{t \in [0,T]} \left\| x_t^0 \right\|^2 \leq \left( C \mathbb{E} \|x_0\|^2 + C \right) \exp(C).$$

As $x_0$ is square-integrable by assumption, therefore we have shown that $\mathbb{E} \sup_{t \in [0,T]} \left\| x_t^0 \right\|^2$ is bounded.

(2) We then show that $\mathbb{E} \sup_{t \in [0,T]} \left\| \partial_\varepsilon x_t^0 \right\|^2$ is also bounded.

From the SDE (39), as we have derived that

$$\partial_\varepsilon x_t^0 = \int_0^t \frac{\partial g_k}{\partial x} \left( x_\tau^0, \tau \right) \partial_\varepsilon x_\tau^0 + \eta_k \left( x_\tau^0, \tau \right) dB_\tau^k,$$

then we have

$$\mathbb{E} \sup_{t \in [0,\tau]} \left\| \partial_\varepsilon x_t^0 \right\|^2 \leq C \mathbb{E} \sup_{t \in [0,\tau]} \left\| \int_0^t \frac{\partial g_k}{\partial x} \left( x_\tau^0, \tau \right) \partial_\varepsilon x_\tau^0 dB_\tau^k \right\|^2 + C \mathbb{E} \sup_{t \in [0,\tau]} \left\| \int_0^t \eta_k \left( x_\tau^0, \tau \right) dB_\tau^k \right\|^2.$$

For $k = 0$, we have

$$\mathbb{E} \sup_{t\in[0,T]} \left\| \int_0^t \frac{\partial g_0}{\partial x}\left(x_\tau^0, \tau\right) \partial_\varepsilon x_\tau^0 dt \right\|^2 + \mathbb{E} \sup_{t\in[0,T]} \left\| \int_0^t \eta_0\left(x_\tau^0, \tau\right) d\tau \right\|^2, \tag{vi}$$

$$\leq C\, \mathbb{E} \sup_{t\in[0,T]} \int_0^t \left\| \frac{\partial g_0}{\partial x}\left(x_\tau^0, t\right) \right\|^2 \left\| \partial_\varepsilon x_\tau^0 \right\|^2 d\tau + C\mathbb{E} \sup_{t\in[0,T]} \int_0^t \left\| \eta_0\left(x_\tau^0, \tau\right) \right\|^2 d\tau, \tag{vii}$$

$$\leq C\, \mathbb{E} \sup_{s\in[0,T]} \left\| \frac{\partial g_0}{\partial x}\left(x_s^0, s\right) \right\|^2 \sup_{t\in[0,T]} \int_0^t \left\| \partial_\varepsilon x_\tau^0 \right\|^2 d\tau + C\, \mathbb{E} \sup_{t\in[0,T]} \int_0^t C\left(1 + \left\| x_\tau^0 \right\|\right)^2 d\tau,$$

$$\leq C + C\, \mathbb{E} \sup_{t\in[0,T]} \int_0^t \left\| \partial_\varepsilon x_\tau^0 \right\|^2 d\tau + C\, \mathbb{E} \sup_{t\in[0,T]} \int_0^t \left\| x_\tau^0 \right\|^2 d\tau, \tag{viii}$$

$$\leq C + C \int_0^T \mathbb{E} \sup_{s\in[0,\tau]} \left\| \partial_\varepsilon x_s^0 \right\|^2 d\tau + C\, \mathbb{E} \sup_{t\in[0,T]} \left\| x_t^0 \right\|^2,$$

where to get to (vi), we used Jensen's inequality; for (vii), we used the linear growth assumption an $\eta_0$, then we obtain (viii) by as derivatives of function $g_0$ are bounded by assumption.
Similarly, for $k = 1, ..., m$,

$$C\, \mathbb{E} \sup_{t\in[0,T]} \left\| \int_0^t \frac{\partial g_1}{\partial x^i}\left(x_\tau^0, \tau\right) \partial_\varepsilon x_\tau^0 dB_\tau \right\|^2 + C\, \mathbb{E} \sup_{t\in[0,T]} \left\| \int_0^t \eta_1\left(x_\tau^0, \tau\right) dB_\tau \right\|^2,$$

$$\leq C\, \mathbb{E} \int_0^T \left\| \frac{\partial g_1}{\partial x}\left(x_\tau^0, \tau\right) \right\|^2 \left\| \partial_\varepsilon x_\tau^0 \right\|^2 d\tau + C\, \mathbb{E} \int_0^T \left\| \eta_1\left(x_\tau^0, \tau\right) \right\|^2 d\tau, \tag{ix}$$

$$\leq C + C \int_0^T \mathbb{E} \sup_{s\in[0,\tau]} ||\partial_\varepsilon x_s^0||^2 d\tau + C\, \mathbb{E} \sup_{t\in[0,T]} ||x_t^0||^2, \tag{x}$$

where we obtain (ix) by the Burkholder-Davis-Gundy inequality and (x) by following similar steps as have shown in (vii) and (viii).
We are now ready to sum up each term to acquire a new inequality:

$$\mathbb{E} \sup_{t\in[0,T]} \left\| \partial_\varepsilon x_t^0 \right\|^2 \leq C + C\, \mathbb{E} \sup_{t\in[0,T]} \left\| x_t^0 \right\|^2 + C \int_0^T \mathbb{E} \sup_{s\in[0,\tau]} \left\| \partial_\varepsilon x_s^0 \right\|^2 d\tau.$$

By Gronwall's lemma, we have that

$$\mathbb{E} \sup_{t\in[0,T]} \left\| \partial_\varepsilon x_t^0 \right\|^2 \leq \left( C + C\, \mathbb{E} \sup_{t\in[0,T]} \left\| x_t^0 \right\|^2 \right) \exp(C).$$

As it is previously shown that $\mathbb{E} \sup_{t\in[0,\tau]} \left\| x^\circ(t) \right\|^2$ is bounded, it is clear that $\mathbb{E} \sup_{t\in[0,T]} \left\| \partial_\varepsilon x_t^0 \right\|^2$ is bounded too.

(3) From similar steps, one can also show that $\mathbb{E} \sup_{t\in[0,T]} \left\| \partial_\varepsilon^2 x_t^0 \right\|^2$ is bounded. $\qquad\square$

*Step 3*: Having shown that $x_t^0, \partial_\varepsilon x_t^0, \partial_\varepsilon^2 x_t^0$ are bounded, we proceed to bound the remainder term by proving the following lemma.

**Lemma B.7.** *For a given $\varepsilon \in \mathbb{R}$, let*

$$\mathcal{R}^\varepsilon := (t, \omega) \mapsto \frac{1}{\varepsilon^3} \left( x^\varepsilon(t, \omega) - x^0(t, \omega) - \varepsilon \partial_\varepsilon x^0(t, \omega) - \varepsilon^2 \partial_\varepsilon^2 x^0(t, \omega) \right),$$

*where the stochastic process $x_t^\varepsilon$ is the solution to Equation (32). Then it holds true that*

$$\mathbb{E} \sup_{t\in[0,T]} \left\| \mathcal{R}^\varepsilon(t) \right\|^2 \text{ is bounded.}$$

*Proof.* The main strategy of this proof is to first rewrite $\varepsilon^3 \mathcal{R}^\varepsilon$ as the sum of some simpler terms and then to bound each term. To simplify the notation, we denote $\tilde{x}_t^\varepsilon$ as $x_t^0 + \varepsilon \partial_\varepsilon x_t^0 + \varepsilon^2 \partial_\varepsilon^2 x_t^0$. For $k = 0, .., n$, we define the following terms:

$$\theta_k(t) := \int_0^t g_k\left(x_\tau^\varepsilon, \tau\right) - g_k\left(\tilde{x}_\tau^\varepsilon, \tau\right) dB_\tau^k,$$

$$\varphi_k(t) := \int_0^t g_k\left(\tilde{x}_\tau^\varepsilon, \tau\right) - g_k\left(x_\tau^0, \tau\right) - \varepsilon \frac{\partial g_k}{\partial x}\left(x_\tau^0, \tau\right) \partial_\varepsilon x_\tau^0 - \varepsilon^2 \Psi_k\left(\partial_\varepsilon x_\tau^0, x_\tau^0, \tau\right) - \varepsilon^2 \frac{\partial g_k}{\partial x^i}\left(x_\tau^0, \tau\right) \partial_\varepsilon^2 x_\tau^0 dB_\tau^k,$$

$$\sigma_k(t) := -\varepsilon \int_0^t \eta_k\left(x_\tau^0, \tau\right) + 2\varepsilon \frac{\partial \eta}{\partial x}\left(x_\tau^0, \tau\right) \partial_\varepsilon x_\tau^0 dB_\tau^k.$$

656    Hence, we have $\varepsilon^3 \mathcal{R}^\varepsilon(t) = \sum_{k=0}^1 \theta_k(t) + \varphi_k(t) + \sigma_k(t)$.

657    For $\theta_k(t)$, we have

$$\mathbb{E} \sup_{t \in [0,T]} \|\theta_k(t)\|^2 \leq C \mathbb{E} \sup_{t \in [0,T]} \int_0^t \left\|g_k\left(x_\varphi^\varepsilon, e\right) - g_k\left(\tilde{x}_\varphi^\varepsilon, \tau\right)\right\|^2 d\tau, \tag{i}$$

$$\leq C \int_0^T \mathbb{E} \sup_{t \in [0, tau]} \|x_t^\varepsilon - \tilde{x}_t^\varepsilon\|^2 d\tau, \tag{ii}$$

$$\leq C \int_0^T \mathbb{E} \sup_{t \in [0,\tau]} \|\mathcal{R}^\varepsilon(t)\|^2 d\tau, , \tag{iii}$$

658    where to obtain (i) we used Jensen's inequality when $k = 0$ and by the Burkholder-Davis-Gundy
659    inequality when $k = 1$, used the Lipschitz condition of $g_k$ to obtain (ii), and for (iii), it is because
660    $\varepsilon^3 \mathcal{R}^\varepsilon(t) = \tilde{x}_t^\varepsilon - x_t^\varepsilon$.
661    We note that from Taylor's theorem, for any $s \in [0,t]$, $k = 0, 1$, there exists some $\varepsilon_s \in (0, \varepsilon)$ s.t.

$$g_k\left(\tilde{x}_s^\varepsilon, s\right) - g_k\left(x_s^0, s\right) - \varepsilon \frac{\partial g_k}{\partial x}\left(x_s^0, s\right) \partial_\varepsilon x_s^0 = \varepsilon^2 \frac{\partial g_k}{\partial x}\left(\tilde{x}_s^{\varepsilon_s}\right) \partial_\varepsilon^2 x_s^0 + \varepsilon^2 \Psi\left(\partial_\varepsilon x_s^0, \tilde{x}_s^{\varepsilon_s}, s\right). \tag{42}$$

662    For $\varphi_k(t)$, we have

$$\mathbb{E} \sup_{t \in [0,T]} \|\varphi_k(t)\|^2$$

$$\leq C \mathbb{E} \sup_{t \in [0,T]} \int_0^t \|\frac{\partial g_k}{\partial x}\left(\tilde{x}_s^{\varepsilon_s}\right) \partial_\varepsilon^2 x_s^0 + \Psi_k\left(\partial_\varepsilon x_s^0, \tilde{x}_s^{\varepsilon_s}, s\right) - \frac{\partial g_k}{\partial x}\left(x_s^0\right) \partial_\varepsilon^2 x_s^0 - \Psi_k\left(\partial_\varepsilon x_s^0, x_s^0, s\right) \|^2 ds, \tag{iv}$$

$$\leq C \mathbb{E} \sup_{t \in [0,T]} \int_0^t \left\|\frac{\partial g_k}{\partial x}\left(\tilde{x}_s^{\varepsilon_s}\right) - \frac{\partial g_k}{\partial x}\left(x_s^0\right)\right\|^2 \left\|\partial_\varepsilon^2 x_s^0\right\|^2 + \left\|\Psi_k\left(\partial_\varepsilon x_s^0, \tilde{x}_s, s\right) - \Psi_k\left(\partial_\varepsilon x_s^0, x_s^0, s\right)\right\|^2 ds, \tag{v}$$

$$\leq C \mathbb{E} \sup_{t \in [0,T]} \int_0^t \left\|\tilde{x}_s^{\varepsilon_s} - x_s^0\right\|^2 \left(C + \left\|\partial_\varepsilon^2 x_s^0\right\|^2\right) ds, \tag{vi}$$

$$\leq C \mathbb{E} \sup_{t \in [0,T]} \int_0^t \left\|\varepsilon \partial_\varepsilon x_s^0 + \varepsilon^2 \partial_\varepsilon^2 x_s^0\right\|^2 \left(C + \left\|\partial_\varepsilon^2 x_s^0\right\|^2\right) ds,$$

$$\leq C \left(\mathbb{E} \sup_{t \in [0,T]} \left\|\partial_\varepsilon x_s^0\right\|^2\right) + \mathbb{E} \sup_{t \in [0,T]} \left\|\partial_\varepsilon^2 x_s^0\right\|^2\right)\right) \left(C + \mathbb{E} \sup_{t \in [0,T]} \left\|\partial_\varepsilon^2 x_s^0\right\|^2\right), \tag{vii}$$

663    where for (iv), we used Equation (42) and Jensen's inequality for $k = 0$ and the Burkholder-Davis-
664    Gundy inequality for $k = 1$; to obtain (v), we applied Jensen's equality; we then derived (vi) from
665    the Lipschitz conditions of $g_k$ and $\Psi_k$; and finally another application of Jensen's inequality gives
666    (vii) which is bounded as a result from the Lemma B.6.
667

668    For $\sigma_k(t)$,

$$\sup_{t\in[0,T]}\|\sigma_0(t)\|^2 \le C\,\varepsilon \int_0^T \mathbb{E}\sup_{s\in[0,t]}\|\eta_k\left(x_s^0,s\right)\|^2 + C\mathbb{E}\sup_{s\in[0,t]}\left\|\frac{\partial\eta_k}{\partial x}\left(x_s^0,s\right)\right\|^2 \|\partial_\varepsilon x_s^0\|^2\,dt, \quad \text{(ix)}$$

$$\le C\int_0^T C\left(1+\mathbb{E}\sup_{s\in[0,t]}\|x_s^0\|^2\right) + C\mathbb{E}\sup_{t\in[0,T]}\left\|\frac{\partial\eta_k}{\partial x}\left(x_t^0,t\right)\right\|^2 \int_0^T \mathbb{E}\sup_{s\in[0,t]}\|\partial_\varepsilon x_s^0\|^2\,dt,$$
$$\text{(x)}$$

$$\le c + C\,\mathbb{E}\sup_t \in[0,T]\,\|x_s^0\|^2 + C\,\mathbb{E}\sup_{t\in[0,T]}\left\|\frac{\partial\eta}{\partial x}\left(x_t^0,t\right)\right\|^2 \mathbb{E}\sup_{t\in[0,T]}\|\partial_\varepsilon x_t^0\|^2,$$
$$\text{(xi)}$$

669    where we obtained (ix) by Jensen's inequality when $k=0$ and by Burkholder-Davis-Gundy inequality
670    when $k=1$, and (x) by the linear growth assumption on $\eta_k$; one can see that (xi) is bounded by
671    recalling the Lemma B.6 and the assumption that $\eta_k$ has bounded derivatives.

672    Hence, by Jensen's inequality and Gronwall's lemma, we have

$$\mathbb{E}\sup_{t\in[0,T]}\|\mathcal{R}^\varepsilon(t)\|^2 \le C\sum_{k=0}^K \mathbb{E}\sup_{t\in[0,T]}\|\theta_k(t)\|^2 + \mathbb{E}\sup_{t\in[0,T]}\|\varphi_k(t)\|^2 + \mathbb{E}\sup_{t\in[0,T]}\|\sigma_k(t)\|^2,$$

$$\le C + C\int_0^T \mathbb{E}\sup_{t\in[0,\tau]}\|\mathcal{R}^\varepsilon(t)\|^2\,d\tau,$$

$$\le C\exp\left(C\right).$$

673    Therefore, $\mathbb{E}\sup\|\mathcal{R}^\varepsilon(t)\|^2$ is bounded.

674    $\square$

675    Finally, it is now straightforward to show Equation (33) by applying a second-order Taylor expansion
676    on $f\left(x_t^0 + \varepsilon\partial_\varepsilon x_t^0 + \varepsilon^2\partial_\varepsilon^2 x_t^0 + \varepsilon^3 R^\varepsilon(t)\right)$.

677    $\square$

678    We are now ready to show Theorem 3.7. One notes that Corollary 3.8 directly follows from the result
679    too.

680    *Proof.* (*Theorem 3.7*) From Proposition B.5, it is noteworthy to point out that the derived SDEs (34)
681    for $\partial_\varepsilon x_t^0$ and $\partial_\varepsilon^2 x_t^0$ are vector-valued general linear SDEs. With some steps of derivations, one can
682    express the solutions as:

$$\partial_\varepsilon x_t^0 = \Phi_t \int_0^t \Phi_s^{-1}\left(\eta_0(x_s^0,s) - \sum_{k=1}^m \frac{\partial g_k}{\partial x}(x_s^0,s)\eta_k(x_s^0,s)\right)ds + \Phi_t \int_0^t \Phi_s^{-1}\eta_k(x_s^0,s)dB_s^k \quad \text{(a)}$$

$$\partial_\varepsilon^2 x_t^0 = \Phi_t \int_0^t \Phi_s^{-1}\left(\Psi_0(x_s^0,\partial_\varepsilon x_s^0,s) + 2\frac{\partial\eta_0}{\partial x}(x_s^0,s)\partial_\varepsilon x_s^0\right.$$

$$\left. - \sum_{k=1}^m \frac{\partial g_k}{\partial x}(x_s^0,s)\left(\Psi_k(x_s^0,\partial_\varepsilon x_s^0,s) + 2\frac{\partial\eta_k}{\partial x}(x_s^0,s)\partial_\varepsilon x_s^0\right)\right)ds,$$

$$+ \Phi_t \int_0^t \Phi_s^{-1}\sum_{k=1}^m \left(\Psi_k(x_s^0,\partial_\varepsilon x_s^0,s) + 2\frac{\partial\eta_k}{\partial x}(x_s^0,s)\partial_\varepsilon x_s^0\right)dB_s^k, \quad \text{(b)}$$

683    where $n\times n$ matrix $\Phi_t$ is the fundamental matrix of the corresponding homogeneous equation:

$$d\Phi_t = \frac{\partial g_k}{\partial x}(x_t^0,t)\,\Phi_t\,dB_t^k, \tag{43}$$

684    with initial value

$$\Phi(0) = I. \tag{44}$$

It is worthy to note that the fundamental matrix $\Phi_t$ is non-deterministic and when $\frac{\partial g_i}{\partial x}$ and $\frac{\partial g_j}{\partial x}$ commutes, $\Phi_t$ has explicit solution

$$\Phi_t = \exp\left( \int_0^t \frac{\partial g_k}{\partial x}(x_s^0, s)dB_s^k - \frac{1}{2}\int_0^t \frac{\partial g_k}{\partial x}(x_s^0, s)\frac{\partial g_k}{\partial x}(x_s^0, s)^\top ds \right). \tag{45}$$

Having obtained the explicit solutions, one can plug in corresponding terms and obtain the results of *Theorem 3.7*) after a Taylor expansion of the loss function $\mathcal{L}$. $\qquad\square$

## C   Error Accumulation During the Inference Phase and its Effects to Value Functions

**Theorem C.1.** *(Error accumulation due to initial representation error )*

*Let $\delta := \mathbb{E}\left\|\varepsilon\right\|$ and $d_\varepsilon := \mathbb{E}\sup_{t\in[0,T]}\left\|h_t^\varepsilon - h_t^0\right\|^2 + \left\|\tilde{z}_t^\varepsilon - \tilde{z}_t^0\right\|^2$. It holds that as $\delta \to 0$,*

$$d_\varepsilon \leq \delta\, C\left(\mathcal{J}_0 + \mathcal{J}_1\right) + \delta^2\, C\left(\exp\left(\mathcal{H}_0\left(\mathcal{J}_0 + \mathcal{J}_1\right)\right) + \exp\left(\mathcal{H}_1\left(\mathcal{J}_0 + \mathcal{J}_1\right)\right)\right) + \mathcal{O}(\delta^3), \tag{46}$$

*where*

$$\mathcal{J}_0 = \exp\left(\mathcal{F}_h + \mathcal{F}_z + \mathcal{P}_h\right),\ \mathcal{J}_1 = \exp\left(\bar{\mathcal{P}}_h\right),$$

$$\mathcal{H}_0 = \mathcal{F}_{hh} + \mathcal{F}_{hz} + \mathcal{F}_{zh} + \mathcal{F}_{zz} + \mathcal{P}_{hh},\ \mathcal{H}_1 = \bar{\mathcal{P}}_{hh}$$

$$\mathcal{F}_h = C\,\mathbb{E}\sup_{t\in[0,T]}\left\|\frac{\partial f}{\partial h} + \frac{\partial f}{\partial a}\partial_h\rho\right\|_F^2,\quad \mathcal{F}_z = C\,\mathbb{E}\sup_{t\in[0,T]}\left\|\frac{\partial f}{\partial z} + \frac{\partial f}{\partial a}\partial_z\rho\right\|_F^2,$$

$$\mathcal{P}_h = C\,\mathbb{E}\sup_{t\in[0,T]}\left\|\frac{\partial p}{\partial h}\right\|_F^2,\ \bar{\mathcal{P}}_h = C\,\mathbb{E}\sup_{t\in[0,T]}\left\|\frac{\partial \bar{p}}{\partial h}\right\|_F^2,$$

$$\mathcal{F}_{hh} = C\,\mathbb{E}\sup_{t\in[0,T]}\left\|\frac{\partial^2 f}{\partial h^2} + \frac{\partial^2 f}{\partial h\partial a}\partial_h\rho + \frac{\partial f}{\partial a}\partial_{hh}^2\rho\right\|_F^2,$$

$$\mathcal{F}_{hz} = C\,\mathbb{E}\sup_{t\in[0,T]}\left\|\frac{\partial^2 f}{\partial h\partial z} + \frac{\partial^2 f}{\partial z\partial a}\partial_h\rho + \frac{\partial f}{\partial a}\partial_{zh}^2\rho\right\|_F^2,$$

$$\mathcal{F}_{zh} = C\,\mathbb{E}\sup_{t\in[0,T]}\left\|\frac{\partial^2 f}{\partial h\partial z} + \frac{\partial^2 f}{\partial h\partial a}\partial_z\rho + \frac{\partial f}{\partial a}\partial_{hz}^2\rho\right\|_F^2,$$

$$\mathcal{F}_{zz} = C\,\mathbb{E}\sup_{t\in[0,T]}\left\|\frac{\partial^2 f}{\partial z^2} + \frac{\partial^2 f}{\partial z\partial a}\partial_z\rho + \frac{\partial f}{\partial a}\partial_{zz}^2\rho\right\|_F^2,$$

$$\mathcal{P}_{hh} = C\,\mathbb{E}\sup_{t\in[0,T]}\left\|\frac{\partial^2 p}{\partial h^2}\right\|_F^2,\ \bar{\mathcal{P}}_{hh} = C\,\mathbb{E}\sup_{t\in[0,T]}\left\|\frac{\partial^2 \bar{p}}{\partial h^2}\right\|_F^2,$$

*where for brevity, when functions always have inputs $(\tilde{z}_t^0, h_t^0, t)$, we adopt the shorthand to write, for example, $f(\tilde{z}_t^0, h_t^0, t)$ as $f$.*

Before proving the main result C.1, we first show the general case of perturbation in initial values. Consider the following general system with noise at the initial value:

$$dx_t = g_0\left(x_t, t\right)dt + g_k\left(x_t, t\right)dB_t^k, \tag{47}$$

$$x(0) = x_0 + \varepsilon, \tag{48}$$

where the initial perturbation $\varepsilon \in \mathbb{R}^n \times \Omega$. As $g_k$ are $\mathcal{C}_g^{2,\alpha}$ functions, by the classical result on the existence and the uniqueness of solution to SDE, there exists a unique solution to Equation (47), denoted as $x_t^\varepsilon$ or $x^\varepsilon(t)$.

To simplify the notation, we write $\partial_i x_t^\varepsilon := \frac{\partial x^\varepsilon(t)}{\partial x^i}, \partial_{ij}^2 x_t^\varepsilon = \frac{\partial^2 x_t^\varepsilon}{\partial x^i\partial x^j}$, for $i, j = 1, \ldots, n$ that are, respectively, the first and second-order derivatives of the solution $x^\varepsilon(t)$ w.r.t. the changes in the corresponding coordinates of the initial value. When $\varepsilon = 0 \in \mathbb{R}^n$, we denote the solutions to Equation (47) as $x_t^0$ with its first and second derivatives $\partial_i x_t^0, \partial_{ij}^2 x_t^0$, respectively.

**Proposition C.2.** *Let $\delta := \mathbb{E}\left\|\varepsilon\right\|$, it holds that*

$$\mathbb{E}\sup_{t\in[0,T]}\left\|x_t^\varepsilon - x_t^0\right\|^2 \leq \sum_{k=0,1} C\,\delta\left(C\,\mathbb{E}\sup_{t\in[0,T]}\left\|\frac{\partial g_k}{\partial x}(x_t^0, t)\right\|_F^2\right) \tag{49}$$

$$+ C\,\delta^2 \exp\left(C\,\mathbb{E}\sup_{t\in[0,T]}\left\|\frac{\partial^2 g_k}{\partial x^2}(x_t^0, t)\right\|_F^2 \sum_{\bar{k}=0,1}\exp\left(C\,\mathbb{E}\sup_{t\in[0,T]}\left\|\frac{\partial g_{\bar{k}}}{\partial x}(x_t^0, t)\right\|_F^2\right)\right) + \mathcal{O}(\delta^3),$$

$$\tag{50}$$

*as $\delta \to 0$.*

*Proof.* Similar to the previous section, for notational convenience, we write $t$ as $B_t^0$ and employs Einstein summation notation. Hence, Equation (47) can be shorten as

$$dx_t = g_k(x_t, t) \, dB_t^k, \tag{51}$$

with initial values $x(0) = x_0 + \varepsilon$.

To begin, we find the SDEs that characterize $\partial_i x_t^\varepsilon$ and $\partial_{ij}^2 x_t^\varepsilon$, for $i, j = 1, ..., n$.

For $\partial_i x_t^\varepsilon$, we apply Theorem 3.1 from Section 2 in [17] on Equation (51) and $\partial_i x_t^\varepsilon$ satisfy the following SDE with probability 1,

$$d\partial_i x_t^\varepsilon = \frac{\partial g_k}{\partial x}(x_t^\varepsilon, t) \, \partial_i x_t^\varepsilon dB_t^k \tag{52}$$

with initial value $\partial_i x_0^\varepsilon$ to be the unit vector $e_i = (0, 0, \ldots, 1, \ldots, 0)$ that is all zeros except one in the $i^{\text{th}}$ coordinate.

For $\partial_{ij}^2 x_t^\varepsilon$, we again apply Theorem 3.1 from Section 2 in [17] on the SDE (52) and obtain that $\partial_{ij}^2 x_b^\varepsilon$ satisfy the following SDE with probability 1,

$$d\partial_{ij}^2 x_t^\varepsilon = \Psi_k(x_t^\varepsilon, \partial_i x_t^\varepsilon, t) \, \partial_{ij}^2 x_t^\varepsilon dB_t^k, \tag{53}$$

with the initial value $\partial_{ij} x^\varepsilon(0) = e_j$, where

$$\Psi_k : \mathbb{R}^d \times \mathbb{R}^d \times [0, T] \to \mathbb{R}^{d \times d}, \ (x, \partial_i x, t) \mapsto \left( \frac{\partial^2 g_k^l}{\partial x^u \partial x^v}(x_t^\varepsilon, t) \right)_{l,u,v} \partial_i x^v.$$

For the next step, we show that with probability 1, the following holds

$$x_t^\varepsilon = x_t^0 + \varepsilon^i \, \partial_i x_t^0 + \frac{1}{2} \varepsilon^i \varepsilon^j \, \partial_{ij}^2 x_t^0 + O(\varepsilon^3), \tag{54}$$

as $\|\varepsilon\| \to 0$.

One can follow the similar steps of proofs for Lemma (B.6) and (B.7) in the previous section to show that $\mathbb{E} \sup_{t \in [0,T]} \|x_t^0\|^2$, $\mathbb{E} \sup_{t \in [0,T]} \|\partial_i x_t^0\|^2$, $\mathbb{E} \sup_{t \in [0,T]} \|\partial_{ij}^2 x_t^0\|^2$ and the remainder term are bounded. Hence, Equation (54) holds with probability 1.

Indeed, for $\mathbb{E} \sup_{t \in [0,T]} \|\partial_i x_t^0\|^2$, it holds that

$$\mathbb{E} \sup_{t \in [0,T]} \|\partial_i x_t^0\|^2 \le C \|e_i\|^2 + \sum_{k=0,1} \mathbb{E} \sup_{t \in [0,T]} C \int_0^t \left\| \frac{\partial g_k}{\partial x}(x_s^0, s) \right\|_F^2 \|\partial_i x_s\|^2 ds \tag{55}$$

$$\le \sum_{k=0,1} C \exp \left( C \mathbb{E} \sup_{t \in [0,T]} \left\| \frac{\partial g_k}{\partial x}(x_t^0, t) \right\|_F^2 \right). \tag{56}$$

Similarly, for $\mathbb{E} \sup_{t \in [0,T]} \|\partial_{ij}^2 x_t^0\|^2$, it holds that

$$\mathbb{E} \sup_{t \in [0,T]} \|\partial_{ij}^2 x_t^0\|^2 \le C \|e_i\|^2 + \sum_{k=0,1} \mathbb{E} \sup_{t \in [0,T]} C \int_0^t \left\| \frac{\partial^2 g_k}{\partial x^2}(x_s^0, s) \right\|_F^2 \|\partial_i x_s^0\|^2 \|\partial_{ij}^2 x_s^0\|^2 ds$$
$$\tag{57}$$

$$\le C \sum_{k=0}^1 \exp \left( C \mathbb{E} \sup_{t \in [0,T]} \left\| \frac{\partial^2 g_k}{\partial x^2}(x_t^0, t) \right\|_F^2 \|\partial_i x_t^0\|^2 \right) \tag{58}$$

$$\le C \sum_{k=0,1} \exp \left( C \mathbb{E} \sup_{t \in [0,T]} \left\| \frac{\partial^2 g_k}{\partial x^2}(x_t^0, t) \right\|_F^2 \exp \left( C \mathbb{E} \sup_{t \in [0,T]} \left\| \frac{\partial g_k}{\partial x}(x_t^0, t) \right\|_F^2 \right) \right). \tag{59}$$

Therefore, we could obtain the proposition by applying Jensen's inequality to Equation (54) and plugging with 56 and 57. $\qquad\square$

729 Now we are ready to prove Theorem C.1. We note that one could then obtain Corollary 4.2 without
730 much more effort by a standard application of Taylor's theorem.

731 *Proof.* (Proof for Theorem C.1)

732 At $(h_t, \tilde{z}_t, \pi(h_t, \tilde{z}_t))$, where the local optimal policy $\pi(h_t, \tilde{z}_t)$, denoted as $a_t^*$, there exists an open
733 neighborhood $V \subseteq \mathcal{A}$ of $a_t^*$ such that $a_t^*$ is the local maximizer for $Q(h_t, \tilde{z}_t, \cdot)$ by definition.
734 Then, $\frac{\partial Q}{\partial a}(h_t, \tilde{z}_t, a_t^*) = 0$, and $\frac{\partial^2 Q}{\partial a^2}(h_t, \tilde{z}_t, a)$ is negative definite. As $\frac{\partial^2 Q}{\partial a^2}$ is non-degenerate in the
735 neighborhood $V$, by the implicit function theorem, there exists a neighborhood $U \times V$ of $(h_t, \tilde{z}_t, a_t^*)$
736 such that there exists a $\mathcal{C}^2$ map $\rho : U \to V$ such that $\frac{\partial Q}{\partial a}(h, \tilde{z}, \rho(h, \tilde{z})) = 0$ and $\rho(h, \tilde{z})$ is the
737 local maximizer of $Q(h, \tilde{z}, \cdot)$ for any $h, \tilde{z} \in U$. Furthermore, we have that $\partial_h \rho = -\frac{\partial^2 Q}{\partial a^2}^{-1} \frac{\partial^2 Q}{\partial a \partial h}$.
738 Similarly, other first-terms and second-order terms $\partial_z \rho, \partial_{zz}^2 \rho, \partial_{zh}^2 \rho, \partial_{hz}^2 \rho, \partial_{hh}^2 \rho$ can be explicitly
739 expressed without much additional effort (e.g., in [28], [3]).

740 The rest of the proof is easy to see after plugging in the corresponding terms from Proposition
741 C.2. □

# D    Experimental Details

In this section, we provide additional details and results beyond thoese in the main paper.

## D.1    Model Implementation and Training

Our baseline is based on the DreamerV2 Tensorflow implementation. Our theoretical and empirical results should not matter on the choice of specific version; so we chose DreamerV2 as its codebase implementation is simpler than V3. We incorporated a computationally efficient approximation of the Jacobian norm for the sequence model, as detailed in [18], using a single projection. During our experiments, all models were trained using the default hyperparameters (see Table 5) for the MuJoCo tasks. The training was conducted on an NVIDIA A100 and a GTX 4090, with each session lasting less than 15 hours.

| Hyperparameter | Value |
|---|---|
| eval_every | 1e4 |
| prefill | 1000 |
| train_every | 5 |
| rssm.hidden | 200 |
| rssm.deter | 200 |
| model_opt.lr | 3e-4 |
| actor_opt.lr | 8e-5 |
| replay_capacity | 2e6 |
| dataset_batch | 16 |
| precision | 16 |
| clip_rewards | tanh |
| expl_behavior | greedy |
| encoder_cnn_depth | 48 |
| decoder_cnn_depth | 48 |
| loss_scales_kl | 1.0 |
| discount | 0.99 |
| jac_lambda | 0.01 |

Table 5: Hyperparameters for DreamerV2 model.

## D.2 Additional Results on Generalization on Perturbed States

In this experiment, we investigated the effectiveness of Jacobian regularization in model trained against a baseline during the inference phase with perturbed state images. We consider three types of perturbations: (1) Gaussian noise across the full image, denoted as $\mathcal{N}(\mu_1, \sigma_1^2)$ ; (2) rotation; and (3) noise applied to a percentage of the image, $\mathcal{N}(\mu_2, \sigma_2^2)$. (In Walker task, $\mu_1 = \mu_2 = 0.5, \sigma_2^2 = 0.15$; in Quadruped task, $\mu_1 = 0, \mu_2 = 0.05, \sigma_2^2 = 0.2$.) In each case of perturbations, we examine a collection of noise levels: (1) variance $\sigma^2$ from 0.05 to 0.55; (2) rotation degree $\alpha$ 20 and 30; and (3) masked image percentage $\beta\%$ from 25 to 75.

## D.3 Walker Task

| $\beta\%$ mask, $\mathcal{N}(0.5, 0.15)$ | mean (with Jac.) | stdev (with Jac.) | mean (baseline) | stdev (baseline) |
|---|---|---|---|---|
| 25% | 882.78 | 28.57199976 | 929.778 | 10.13141451 |
| 30% | 878.732 | 40.92085898 | 811.198 | 7.663919934 |
| 35% | 856.32 | 37.56882045 | 799.98 | 29.75286097 |
| 40% | 804.206 | 47.53578989 | 688.382 | 43.21310246 |
| 45% | 822.97 | 80.36907477 | 601.862 | 42.49662057 |
| 50% | 725.812 | 43.87836335 | 583.418 | 76.49237076 |
| 55% | 768.68 | 50.71423045 | 562.574 | 59.88315135 |
| 60% | 730.864 | 23.37324967 | 484.038 | 90.38940234 |
| 65% | 696.936 | 65.26307708 | 516.936 | 41.44549462 |
| 70% | 687.346 | 70.9078686 | 411.922 | 45.85808832 |
| 75% | 685.492 | 63.22171723 | 446.74 | 40.66898799 |

Table 6: *Walker.* Mean and standard deviation of accumulated rewards under masked perturbation of increasing percentage.

| full, $\mathcal{N}(0.5, \sigma^2)$ | mean (with Jac.) | stdev (with Jac.) | mean (baseline) | stdev (baseline) |
|---|---|---|---|---|
| 0.05 | 894.594 | 39.86907737 | 929.778 | 40.91 |
| 0.10 | 922.854 | 27.28533819 | 811.198 | 98.79 |
| 0.15 | 941.512 | 16.47165049 | 799.98 | 106.01 |
| 0.20 | 840.706 | 66.12470628 | 688.382 | 70.78 |
| 0.25 | 811.764 | 75.06276427 | 601.862 | 83.65 |
| 0.30 | 779.504 | 53.29238107 | 583.418 | 173.59 |
| 0.35 | 807.996 | 34.35949621 | 562.574 | 79.30 |
| 0.40 | 751.986 | 85.20137722 | 484.038 | 112.43 |
| 0.45 | 663.578 | 60.18862658 | 516.936 | 90.25 |
| 0.50 | 618.982 | 61.10094983 | 411.922 | 116.94 |
| 0.55 | 578.62 | 64.25840684 | 446.74 | 84.44 |

Table 7: *Walker.* Mean and standard deviation of accumulated rewards under Gaussian perturbation of increasing variance.

| rotation, $\alpha°$ | mean (with Jac.) | stdev (with Jac.) | mean (baseline) | stdev (baseline) |
|---|---|---|---|---|
| 20 | 423.81 | 12.90174678 | 391.65 | 35.33559636 |
| 30 | 226.04 | 23.00445979 | 197.53 | 15.26706914 |

Table 8: *Walker.* Mean and standard deviation of accumulated rewards under rotations.

 **D.4 Quardruped Task**

| $\beta\%$ mask, $\mathcal{N}(0.5, 0.15)$ | mean (with Jac.) | stdev (with Jac.) | mean (baseline) | stdev (baseline) |
|---|---|---|---|---|
| 25% | 393.242 | 41.10002579 | 361.764 | 81.41175179 |
| 30% | 384.11 | 20.70463958 | 333.364 | 101.7413185 |
| 35% | 354.222 | 53.14855379 | 306.972 | 16.02275164 |
| 40% | 329.404 | 39.1193856 | 266.088 | 51.20298351 |
| 45% | 360.662 | 36.86801622 | 281.342 | 47.85950867 |
| 50% | 321.556 | 27.66758085 | 222.222 | 22.0668251 |
| 55% | 300.258 | 31.44931987 | 203.578 | 14.38754218 |
| 60% | 321 | 18.42956321 | 217.98 | 23.81819368 |
| 65% | 304.62 | 20.75493676 | 209.238 | 47.14895407 |
| 70% | 301.166 | 18.2485583 | 193.514 | 60.83781004 |
| 75% | 304.92 | 18.63214963 | 169.58 | 30.83637462 |

Table 9: *Quadruped.* Mean and standard deviation of accumulated rewards under masked perturbation of increasing percentage.

| full, $\mathcal{N}(0, \sigma^2)$ | mean (with Jac.) | stdev (with Jac.) | mean (baseline) | stdev (baseline) |
|---|---|---|---|---|
| 0.10 | 416.258 | 20.87925573 | 326.74 | 40.30425536 |
| 0.15 | 308.218 | 24.26432093 | 214.718 | 15.7782198 |
| 0.20 | 314.29 | 44.73612075 | 218.756 | 35.41520832 |
| 0.25 | 293.02 | 24.29582269 | 190.78 | 26.22250465 |
| 0.30 | 269.778 | 21.83423047 | 207.336 | 39.1071161 |
| 0.35 | 282.046 | 13.55303767 | 217.048 | 29.89589972 |
| 0.40 | 273.814 | 19.81361476 | 190.208 | 59.61166975 |
| 0.45 | 267.18 | 17.5276068 | 195.606 | 18.91137964 |
| 0.50 | 268.838 | 29.45000543 | 194.082 | 26.76677642 |
| 0.55 | 252.54 | 22.516283 | 150.786 | 24.53362855 |

Table 10: *Quadruped.* Mean and standard deviation of accumulated rewards under Gaussian perturbation of increasing variance.

| rotation, $\alpha^\circ$ | mean (with Jac.) | stdev (with Jac.) | mean (baseline) | stdev (baseline) |
|---|---|---|---|---|
| 20 | 787.634 | 101.5974723 | 681.032 | 133.7507948 |
| 30 | 610.526 | 97.74499159 | 389.406 | 61.5997198 |

Table 11: *Quadruped.* Mean and standard deviation of accumulated rewards under rotations.

### D.5 Additional Results on Robustness against Encoder Errors

In this experiment, we evaluate the robustness of model trained with Jacobian regularization against two exogenous error signals (1) zero-drift error with $\mu_t = 0, \sigma_t^2$ ($\sigma_t^2 = 5$ in Walker, $\sigma_t^2 = 0.1$ in Quadruped), and (2) non-zero-drift error with $\mu_t \sim [0, 5], \sigma_t^2 \sim [0, 5]$ uniformly. $\lambda$ weight of Jacobian regularization is $0.01$. In this section, we included plot results of both evaluation and training scores.

### D.5.1 Walker Task

Under the Walker task, Figures 3 and 4 show that model with regularization is significantly less sensitive to perturbations in latent state $z_t$ compared to the baseline model without regularization. This empirical observation supports our theoretical findings in Corollary 3.8, which assert that the impact of latent representation errors on the loss function $\mathcal{L}$ can be effectively controlled by regulating the model's Jacobian norm.

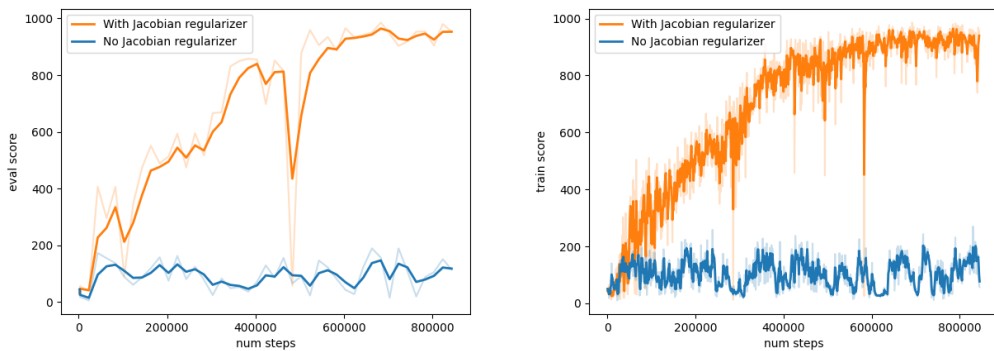

Figure 3: *Walker.* Eval (left) and train scores (right) under latent error process $\mu_t = 0, \sigma_t^2 = 5$
.

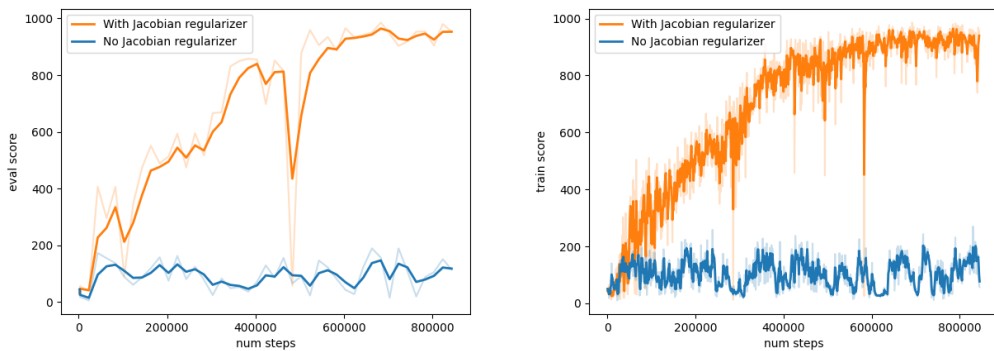

Figure 4: *Walker.* Eval (left) and train scores (right) under latent error process $\mu_t \sim [0, 5], \sigma_t^2 \sim [0, 5]$.

 **D.5.2 Quadruped Task**

 Under the Quadruped task,we initially examined a smaller latent error process ($\mu_t = 0, \sigma_t^2 = 0.1$) and
 observed that the model with Jacobian regularization converged significantly faster, even though the
 adversarial effects on the model without regularization were less severe (Figure 5). When considering
 the more challenging latent error process ($\mu_t \sim [0, 5], \sigma_t^2 \sim [0, 5]$), we noted that the regularized
 model remained significantly less sensitive to perturbations in latent state $z_t$, whereas the baseline
 model struggled to learn (Figure 6). These empirical observations reinforce our theoretical findings
 in Corollary 3.8, demonstrating that regulating the model's Jacobian norm effectively controls the
impact of latent representation errors.

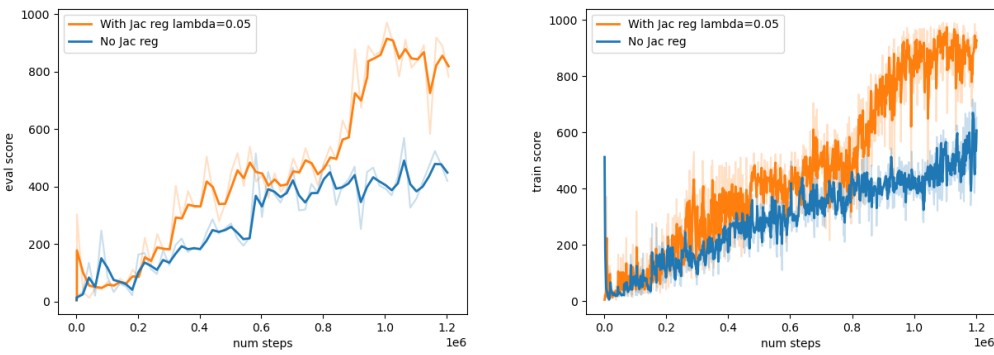

Figure 5: *Quad.* Eval (left) and train scores (right) under latent error process $\mu_t = 0, \sigma_t^2 = 0.1$.

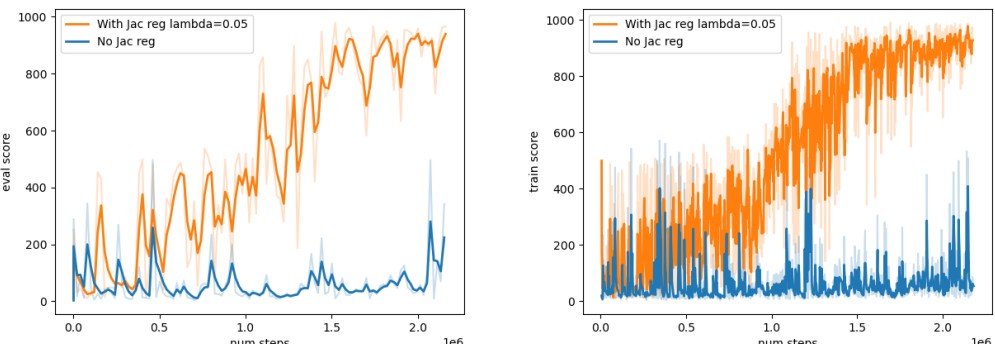

Figure 6: *Quad.* Eval (left) and train scores (right) under latent error process $\mu_t \sim [0, 5], \sigma_t^2 \sim [0, 5]$.

## D.6 Additional Results on Faster convergence on tasks with extended horizon.

In this experiment, we evaluate the efficacy of Jacobian regularization in extended horizon tasks, specifically by increasing the horizon length in MuJoCo Walker from 50 to 100 steps. We tested two regularization weights $\lambda = 0.1$ and $\lambda = 0.05$. Figure 7 demonstrates that models with regularization converge faster, with $\lambda = 0.05$ achieving convergence approximately 100,000 steps ahead of the model without Jacobian regularization. This supports the findings in Theorem 4.1, indicating that regularizing the Jacobian norm can reduce error propagation, especially over longer time horizons.

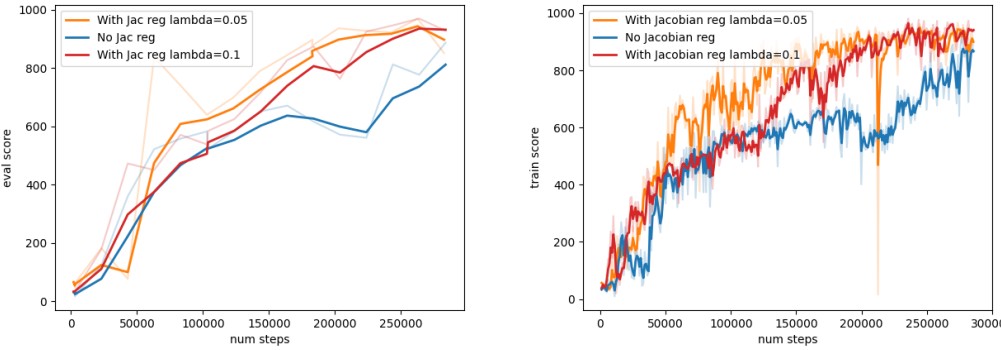

Figure 7: *Extended horizon Walker task.* Eval (left) and train scores (right).

