# OpenReview forum: "Towards Unraveling and Improving Generalization in World Models"
_NeurIPS.cc/2024/Conference — Submitted to NeurIPS 2024_

### Official Review · Reviewer_waKG · 2024-07-04

**Soundness:** 3
**Presentation:** 2
**Contribution:** 2
**Rating:** 6
**Confidence:** 3

**Summary:**

This paper investigates the generalization capabilities of world models in RL, particularly with respect to latent representation errors, which arise when observations are encoded into a low-dimensional latent space. The authors provide a bound on latent representation error when using CNN encoder-decoder architectures. The world model is framed as a stochastic differential equation to characterize the impact of latent representation errors on generalization in terms of either zero or non-zero drift. The authors provide theoretical analysis which shows that these errors can result in implicit regularization in the zero drift case, and propose a Jacobian regularization scheme to tackle the unwanted bias term in the non-zero drift case. Finally, when performing model rollouts for learning a policy, the authors study the effect of these errors on the value function. Experiments on Mujoco tasks demonstrate that the proposed Jacobian regularization enhances robustness to noisy states, reduces the detrimental impact of latent representation errors, and improves convergence speed for longer horizon tasks.

**Strengths:**

- World models are a popular area of research in the RL community, but there is a lack of theoretical understanding. This paper takes one step towards theoretically analyzing the generalization capabilities of world models.
- The analysis of the effect of latent representation error is a novel theoretical contribution, to the best of my knowledge.
- The results in the paper seem mathematically sound and provide useful insights. The empirical results demonstrate that the Jacobian regularization, which naturally arises from the theoretical analysis, is helpful in improving robustness.
- As a very theory-heavy paper, the authors structured the writing such that it makes it easy to follow each individual result (though there is some room for improvement here, see weaknesses).

**Weaknesses:**

While the paper studies a previously unexplored problem, there are some questions about the significance of these findings and the use of drift and diffusion terms to represent the error. Other areas for improvement include explaining the insights from the theoretical analysis more clearly, describing the experimental settings in more detail, and supporting certain claims with more evidence.

- Studying the effect of latent representation error is certainly useful, however, with recent advances in representation learning approaches, one can learn reasonably good representations such that the reconstruction error is negligible. When it comes to model-based RL, a much bigger issue is the compounding model error, which is a result of error in the latent/state dynamics model predictions. A comment from the authors on this aspect would be helpful.
- The decomposition of latent error into drift and diffusion terms seems a bit contrived. It is not clear how the error can be expressed in this form, and what defines the scenarios of zero versus non-zero drift.
- The interpretation that propagation of latent error leads to the model exploring novel states seems somewhat questionable. My understanding is that the erroneous states improve robustness similar to noise injection, but will most likely not be valid states belonging to the state space of the MDP. Some reasonable evidence is required to support this statement.
- The paper presents several results and including some intuitive or low-level explanation for each of those results would greatly improve readability. Additionally, due to the large amount of mathematical notation used throughout the paper, it would be helpful to include a notation table in the appendix for easy reference.
- The experimental setting is not sufficiently clear, especially in the introduction when the authors refer to Table 1. With regards to the perturbations - are they applied to every state in the trajectory? For masking, is the same mask used for every state, or is the mask also sampled randomly? With regards to injecting encoder error - how to interpret the $\mu_t$ and $\sigma_t$ values?

**Questions:**

- I am not sure I fully understand the relation of batch size with the latent representation error in Table 1. Since one would take the mean over the batch size, it should not affect the error magnitude. Is the variation in performance due to stochasticity of the SGD updates? If so, how does it relate to latent representation error? Also, as mentioned in weaknesses, the experimental setting should be clearly explained at this point in the paper.
- What do the drift and diffusion terms corresponding to the latent representation error signify?
- Theorem 3.7 suggests that the regularizing effect of the latent representation error can be attributed to the Hessian of the loss function, which encourages wider minima. As noted by the authors, this term is non-negative only if the loss function is convex. Could the authors comment on how to interpret this result for non-convex loss functions (which is usually the case in deep learning)?
- The analysis in this paper focuses on Dreamer style models which use an RNN to represent the latent dynamics. A parallel line of work, TD-MPC [1], uses an MLP model to predict the next state which is applied recursively. I am interested to know if the authors have any thoughts on the applicability of their analysis to such methods, and if there are any major differences.
- An alternate method to improve robustness to perturbations is by training the model/value function on different augmentations of the states [2]. I understand the limitations of time during the short rebuttal period, but an empirical comparison would significantly enhance the paper. I am also curious to know the authors’ thoughts on the pros/cons of using the Jacobian regularization term over such data augmentation methods.
- How much computational overhead is added by the calculation of the Jacobian regularization term?

[1] Hansen, N.A., Su, H. and Wang, X., 2022, June. Temporal Difference Learning for Model Predictive Control. In *International Conference on Machine Learning* (pp. 8387-8406). PMLR.

[2] Yarats, D., Kostrikov, I. and Fergus, R., 2021, May. Image augmentation is all you need: Regularizing deep reinforcement learning from pixels. In *International conference on learning representations*.

**Limitations:**

There is little discussion on the limitations of the analysis. Some points worth discussing could be the impact of various assumptions when deriving the results, the fact that the analysis is mostly focused on a specific setting - learning from pixels using a CNN encoder and an RNN latent dynamics model, and further investigation of the compounding model error problem.

---

> ### Author Rebuttal · Authors · 2024-08-07
>
> Thank you very much for your helpful feedback on our work! We respond to your questions as follow:
>
> ### **A1. On the analysis of compounding model error**
>
> Thank you for highlighting this important aspect. We agree that compounding model error is critical in world model analysis. During both training and predictive rollouts, even if the latent representation error is small at each time step, its accumulation through the dynamics model's predictions throughout the trajectory is significant. Our results show that in the training phase, the accumulated effects of zero-drift errors can act as implicit regularization (which is somewhat surprising but consistent with noise injection technique used in deep learning [1, 2]), while non-zero drift errors require additional regularization. During predictive rollouts, accumulated errors can lead to divergence if not properly regularized. Importantly, the degree of these effects depends on the dynamics model’s Jacobian. Hence, we propose Jacobian regularization.
>
> Our theoretical results explicitly consider these compounding effects by interpreting world models as SDE dynamical systems to characterize error propagation. For instance, in Theorem 3.7 and Corollary 3.8, the derived regularization terms $\mathcal{R}$ and bias term $\tilde{\mathcal{R}}$ account for the accumulative effects of continuously introduced encoder errors on the sequence model $f$ and transition predictor $p$. This means that the effects of encoder errors from the initial time step to the entire trajectory of state predictions are captured in $\mathcal{R}$ and $\tilde{\mathcal{R}}$.
>
> Regarding the compounding model error due to additional inaccuracies in the latent/state dynamics model predictions, our main results can be generalized to include errors from other model components, such as transition predictor error. For example, one could incorporate error terms $\sigma$ as $(\sigma_{\text{enc}}, 0, \sigma_{\text{pred}}, 0)$ and $\bar{\sigma}$ as $(\bar{\sigma}\_{\text{enc}}, 0, \bar{\sigma}\_{\text{pred}}, 0)$ in Theorem 3.7 and Corollary 4.2.
>
> 1. Alexander Camuto et al., "Explicit Regularisation in Gaussian Noise Injections," Advances in Neural Information Processing Systems 34 (NeurIPS 2020), 2020.
>
> 2. Soon Hoe Lim et al., "Noisy Recurrent Neural Networks," Proceedings of the 35th Conference on Neural Information Processing Systems (NeurIPS 2021), 2021.
>
> ### **A2. On the drift and diffusion components of error (and Q2)**
>
> Thank you for the question. In our case, the decomposition of latent representation error into drift and diffusion terms arises naturally as additive noise to the encoder, which is interpreted as an SDE process with drift and diffusion coefficient functions matching the stochastic design of the world model. In general, since the error process is a stochastic term, it is natural to consider its mean (drift) and variance (diffusion) decomposition, a common approach in stochastic control (see [3] for more details).
>
> The case of zero-drift noise occurs when the learned encoder is unbiased but has variance. In contrast, the case of non-zero-drift noise corresponds to a more general situation where the learned encoder is biased and has variance. These are general settings that occur in world model learning.
>
> 3. Ramon van Handel, "Stochastic Calculus, Filtering, and Stochastic Control: Lecture Notes," Spring 2007.
>
>
> ### **A3. On the interpretation of latent representation error encouraging state exploration**
>
> Thank you for your insightful comment. We respectfully disagree with the reviewer’s assertion that erroneous latent states lead the agent to explore invalid states outside the state space of the MDP. We note that since the agent’s policy in the world model is conditioned on latent states, erroneous latent states $\tilde{z}$ lead the agent to take inaccurate actions $\tilde{a}$. The gap in the predicted value function due to latent representation error is captured in Corollary 4.2.
>
> From the perspective of the task environment, the agent begins at a valid state $s$ and takes the inaccurate action $\tilde{a}$, but will still transition to a valid next state $\mathcal{T}(s, \tilde{a})$. This suboptimality can be interpreted as implicitly encouraging exploration by introducing a stochastic perturbation in the value function. Therefore, the agent remains within the valid state space while experiencing robustness similar to noise injection, as you suggested.
>
> ### **A4. On including low-level intuition for each result and overall readability**
>
> We appreciate your helpful suggestion. We will include a notation table in the appendix for easy reference. We will also provide more intuitive explanations for our theoretical results, expanding on the current ones, including
>
> Theorem 3.7: “The presence of this Hessian-dependent term S, under latent representation error, implies a tendency towards wider minima in the loss landscape…” (page 6, 234-235)
>
> Corollary 3.8: “The presence of $\tilde{\zeta}$ in ,$\tilde{Q}$ and $\tilde{S}$ induces a bias to the loss function with its magnitude dependent on the error level $\epsilon$, since $\tilde{zeta}$ is a non-zero term influenced on the drift term $\sigma$.” (page 7, 260-262)
>
> Theorem 4.1: “... the expected divergence from error accumulation hinges on the expected error magnitude, the Jacobian norms within the latent dynamics model and the horizon length T.” (page 7, 300-302)
>
> We believe these additions will greatly improve the readability and accessibility of our results.
>
> _**Rebuttal will continue in the comments.**_

---

> ### Author Response · Authors · 2024-08-07
> **Rebuttal 2/4 for Reviewer waKG**
>
> ### **A5. On the clarification of experiment settings**
>
> We apologize for the confusion and will clarify further in the experiment section in the appendix. For the batch-size versus robustness experiment in Table 1, the considered perturbation methods are the same as those in the experiment on Jacobian regularization with perturbed states (Table 2 and Appendix D.2).
>
> Regarding to your specific questions:
>
> _—”With regards to the perturbations - are they applied to every state in the trajectory?”_
>
> Yes, the perturbations are applied at every time step. This consistency is maintained, particularly in the robustness experiments with encoder noise, to align with our theoretical results, which studied continuously introduced encoder noise.
>
> _—”For masking, is the same mask used for every state, or is the mask also sampled randomly”_
>
> The masks are sampled randomly for each state, not the same mask for every state.
>
> _—”With regards to injecting encoder error - how to interpret the $\mu_t$ and $\sigma_t$ values?”_
>
> We consider a Gaussian noise process where at each time step, where $\mu_t$ amd $\sigma_t$ denotes mean and variance of the Gaussian process at time $t$ . An explanation is provided in the appendix (see pages 28, line 756, and pages 30, lines 764-765).
>
> In the revised version, with more page space available, we will include further clarification of the experimental settings in Section 5 from the main text to avoid any confusion.
>
> ### **A6. Responses to Questions**
>
> ### Q1: On the gradient estimation errors in Table 1
>
> Thank you for your insightful question.
> We used batch size experiments as a motivating example to introduce the relationship between error and generalization robustness. Batch-induced gradient estimation errors are relatively well-studied, whereas latent representation error is often overlooked. Our theoretical results on the effects of latent representation error interestingly align with the discovered empirical effects on gradient estimation error in the zero-drift case:
>
> -	Zero-mean error: Zero-mean errors, such as gradient estimation error and zero-drift latent regularization errors, have the potential for generalization improvement.
> -	Controlling the error: To harness these generalization gains, the batch size can influence gradient estimation error. When kept within a controlled range, this can enhance generalization—paralleling the control of latent representation error through Jacobian regularization.
>
> In the revision, we will further clarify these points to avoid any confusion and provide a clearer explanation of the experimental settings of Table 1 in Section D in appendix.
>
>
> ### Q3: Interpreting results for non-convex case
>
> Thank you for the interesting question.
>
> In the non-convex case, our most important finding from Corollary 3.8 remains relevant: the additional bias induced by the drift term of latent representation error can be controlled through the model’s Jacobian norm. While non-convex optimization presents more challenges due to the complicated gradient landscape, this control mechanism provides a valuable tool.
>
> For further insights, non-convex optimization remains an open challenge, and we acknowledge the significant complexity of its gradient landscape. We are excited to consider future work that focuses on specific types of non-convex loss functions to extract more detailed insights and refine our understanding of the regularizing effects in these scenarios.
>
> ### Q4: Extending the analysis on TD-MPC
>
> For TD-MPC, which has a deterministic latent dynamics model, our SDE analysis can specialize to TD-MPC by setting all diffusion coefficient functions to zero in formulation equations (5) - (8), thus becoming systems of ODEs, provided that the learned MLP model in TD-MPC satisfies certain continuity assumptions to guarantee the unique existence of solutions.
>
> Regarding the effects of latent representation error in TD-MPC, this would resemble a simplified version of Corollary 3.8 without diffusion terms. The result would similarly introduce a bias to the loss function, with its magnitude dependent on the error level. This effect can be modulated by the MLP’s input-output Jacobian norm. We are excited about future work to develop the theoretical details and experimental results in this context.

---

> ### Author Response · Authors · 2024-08-07
> **Rebuttal 3/4 for Reviewer waKG**
>
> ### Q5: — “An alternate method to improve robustness to perturbations is by training the model/value function on different augmentations of the states”
>
> We agree that training the model/value function on different augmentations of the states is a valuable approach to improving robustness to perturbations. Following your suggestion, we added a new set of data augmentation experiments for comparison. We considered training with state images augmented with randomly-masked Gaussian noises. We evaluated it against 3 different perturbations: 1) modifying gravity constant g in the DMC walker environment 2) adding rotation to input image state 3) adding masked Gaussian noise to input image state.
>
> The experiment results show that under varied gravity g and rotation perturbation, models trained with Jacobian regularization outperform state augmentation. Under masked Gaussian noise perturbation, Jacobian regularization outperforms state augmentation when the noise is more dense (mask with $\beta=70, 60$) and underperforms when $\beta=40, 50$. This may be due to the fact that state augmentation is trained with very similar Gaussian noise. State augmentation works well when the augmentation is consistent with the perturbation in inference, but does not generalize to unseen perturbations. Please see below table and Figure 3 in attached PDF for details.
>
>
> |   Gravity($m/s^2$) g                             |  g = 9.8 (default)| g = 6                   | g = 3                   | g = 1
> |---------------------------------|--------------------------|-------------------------|-------------------------|-------------------------|
> | Aug w. $\mathcal{N}(0.15, 0.1)$ | 847.19 ± 131.85          | 771.34 ± 88.112         | 550.4 ± 75.8            |  390.7 ± 94.28          |
> | Jac Reg ($\lambda = 0.01$)      | **920.24 ± 39.952**      | **906.42 ± 42.664**     | **798.02 ± 95.936**     | **603.88 ± 162.224**    |
>
> | Rotation ($^\circ$) $\alpha$     |  $\alpha = 20$           | $\alpha = 25$             | $\alpha = 30$
> |---------------------------------|--------------------------|-------------------------|-------------------------|
> | Aug w. $\mathcal{N}(0.15, 0.1)$ | 286.63 ± 81.678          | 284.09 ± 59.801         | 213.93 ± 42.44          |
> | Jac Reg ($\lambda = 0.01$)      | **423.81 ± 12.9**        | **301.84 ± 20.26**     | **226.04 ± 23.00**       |
>
> | $\beta\\%$ masked $\mathcal{N}(0.5 ,0.15)$  |  $\beta = 40$            | $\beta = 50$             | $\beta = 60$           | $\beta = 70$
> |-------------------------------------------|--------------------------|-------------------------|-------------------------|-------------------------|
> | Aug w. $\mathcal{N}(0.15, 0.1)$           | **846.76 ± 46.928**      | **767.92 ± 78.256**     | 373.08 ± 64.056     | 247.68 ± 54.576     |
> | Jac Reg ($\lambda = 0.01$)                | 804.21 ± 80.369      | 725.81 ± 50.714     | **730.87 ± 65.263**     | **687.35 ± 63.222**    |
>
> The limitations of the rebuttal period prevent us from conducting a comprehensive empirical comparison at this time (e.g. more augmentation patterns), but we are excited to continue this line of research in our future works.
>
> — “on the pros/cons of using the Jacobian regularization term over such data augmentation methods.”
>
> Regarding the pros and cons of using the Jacobian regularization over data augmentation, we note the following:
>
> Pros:
>
> -   Theoretical guarantees: Jacobian regularization is a principled approach grounded in theoretical results (Theorem 3.7 and Corollary 3.8), providing explicit control over model behavior in response to small error.
>
> -   Less reliant on data diversity: Unlike data augmentation, which heavily relies on diversity and relevance of augmented samples, Jacobian regularization targets the learning dynamics of WM.
>
> -   Less likelihood of overfitting: Jacobian regularization is less prone to overfitting as unlike data augmentation, which can lead to overfitting when overly reliant on certain perturbation patterns rather than general robustness.
>
> -   Bounding trajectory divergence: Jacobian regularization also mitigates error propagation during predictive rollouts (Theorem 4.1), a benefit not achieved by data augmentation.
>
>
> Cons:
>
> -   Uncertainties with large error: The theoretical analysis for Jacobian regularization assumes the latent representation error is small. In cases when the encoder remains poorly learned, data augmentation may provide better model robustness.
>
> -   Computational overhead: computing Jacobian terms can introduce additional overhead
>
>
> ### Q6: On the computational overhead of Jacobian regularization term
>
> For every episode with 500 steps on an A100, training the model with Jacobian regularization took 28.702 seconds compared to 22.199 seconds for training without it (averaged over 20 episodes).

---

> ### Author Response · Authors · 2024-08-07
> **Rebuttal 4/4 for Reviewer waKG**
>
> ### **A7. Response to Limitation**
>
>
> We appreciate you raising this point. In the revision, with more page space, we will include a separate section addressing the limitations of our work. We will highlight that the main limitation of the SDE interpretation of world models (WM) is its restriction to a popular family of world models for theoretical soundness. As mentioned in the paper (lines 123-125), "we consider a popular class of world models, including Dreamer and PlaNet, where $\{z, \tilde{z}, \tilde{s}\}$ have distributions parameterized by neural networks’ outputs, and are Gaussian when the outputs are known."
>
> Additionally, our results on the approximation error of latent representation focus on CNN (and similar) models, and future work is needed to generalize the study to models such as transformers. We are exploring this direction in our future work.

---

> > ### Comment · Reviewer_waKG · 2024-08-11
> >
> > I thank the authors for their detailed responses, which clarify several questions regarding the interpretation of the drift and diffusion coefficients, the exploration of novel states, and the experiment details.
> > Additionally, the comparison with data augmentation methods enhances the empirical results and provides a more well-rounded understanding of the proposed Jacobian regularization method.
> > I increase my score to reflect the changes.

---

> > > ### Author Response · Authors · 2024-08-11
> > >
> > > Thank you for your thoughtful feedback and for considering our responses. We are pleased that our clarifications and the comparison with data augmentation methods were helpful. We greatly appreciate your updated score and the opportunity to improve our work.

---

### Official Review · Reviewer_mgpu · 2024-07-12

**Soundness:** 3
**Presentation:** 3
**Contribution:** 3
**Rating:** 6
**Confidence:** 2

**Summary:**

The paper studies the generalization capability of world models via a stochastic differential equation formulation. They try to understand latent representation errors on generalization, with both zero-drift representation errors and non-zero-drift representation errors. They found that zero drift latent representation errors are implicit regularization and thus bring generalization gain. Jacobian regularization is proposed to enhance training stability and generalization.

**Strengths:**

+ A deep understanding of the generalization of world models via stochastic differential equation formulation;
+ A careful study of the different effects of zero drift and non-zero drift on gn

**Weaknesses:**

+ The unseen images are produced via global/partial Gaussian noises and rotation, which seems more on the robustness side rather than the generalization of unseen images;

**Questions:**

See the Weaknesses part and explain how to extend the analysis to more general cases.

---

> ### Author Rebuttal · Authors · 2024-08-07
>
> Thank you very much for your helpful comment!
>
> - _“The unseen images are produced via global/partial Gaussian noises and rotation, which seems more on the robustness side rather than the generalization of unseen images;”_
>
> Following your valuable feedback, we added a new set of experiments involving unseen dynamics to evaluate the robust generalization of our proposed Jacobian regularization. For Mujoco tasks, we varied the acceleration constants due to gravity (default being 9.8 m/s²). This tests the learned agent’s model and the latent dynamics model’s generalization capacity to unseen dynamics. Our results indeed validate that Jacobian regularization (both $\lambda = 0.01$ and $0.05$) outperforms the baseline. We report the mean and variants of eval returns across 5 eval runs in the following table. Please also see Figure 3 in the attached PDF for a plot of varied g vs eval returns.
>
> | Gravity($m/s^2$) | Baseline          | Jac Reg ($\lambda=0.01$)         | Jac Reg ($\lambda=0.05$)         |
> |------------|--------------------------|-------------------------|-------------------------|
> | $g=1.0$         | 381.14 ± 132.968         | 603.88 ± 162.224        | **668.86 ± 89.072**     |
> | $g=3.0$         | 569.64 ± 65.048          | 798.02 ± 95.936         | **717.22 ± 95.056**     |
> | $g=6.0$         | 750.36 ± 122.248         | **906.42 ± 42.664**        | 830.64 ± 62.848     |
> | $g=9.8$ (default)      | 936.32 ± 29.176          | **920.24 ± 39.952**         | 904.918 ± 34.4944   |
>
>
> We also acknowledge the challenges of setting up experiments for completely unseen/unrelated states and transitions in RL due to the complexity of defining such scenarios comprehensively. We considered local/global Gaussian noise, rotation, and varying acceleration constants due to gravity as perturbations. These were designed to best validate our theoretical findings, specifically Theorem 3.7 and Corollary 3.8, on implicit regularization, which link to favoring model solutions in regions of the loss landscape with improved generalization and robustness. Our approach is consistent with the literature's understanding of robust generalization [1, 2].
>
>
> In the revised experiment section, we will clarify the distinction between robust generalization and more generic generalization to avoid any possible confusion.
>
> 1. Binghui Li et al., "Why Robust Generalization in Deep Learning is Difficult: Perspective of Expressive Power," Proceedings of the 36th Conference on Neural Information Processing Systems (NeurIPS 2022), 2022.
>
> 2. Soon Hoe Lim et al., "Noisy Recurrent Neural Networks," Proceedings of the 35th Conference on Neural Information Processing Systems (NeurIPS 2021), 2021.

---

> > ### Author Response · Authors · 2024-08-11
> >
> > Thank you for your valuable feedback and for taking the time to review our work. We have carefully addressed your comments in our responses.
> >
> > We kindly remind you to review our responses, and we are happy to address any further questions you may have.
> >
> > Thank you once again for your thoughtful insights and for contributing to the improvement of our work.

---

> > ### Comment · Reviewer_mgpu · 2024-08-12
> >
> > The authors clarified my major concern about whether the tackled aspect is robustness or generalization. The added new experiment is very helpful.

---

> > > ### Author Response · Authors · 2024-08-12
> > >
> > > We are glad that the additional experiment and clarification was helpful in addressing your concerns. Thank you very much for your updated score and your valuable insights.

---

### Official Review · Reviewer_pwHm · 2024-07-12

**Soundness:** 3
**Presentation:** 3
**Contribution:** 3
**Rating:** 7
**Confidence:** 4

**Summary:**

This paper explores the generalization capability of world models in reinforcement learning. In particular, they investigate the latent representation error in world models. They show that zero-drift representation error is inherently a regularizer for the learned model functions. On the other hand, they show that the non-zero-drift representation error accumulates errors and Jacobian regularization can be used to alleviate the issue. They demonstrate their proposed approach improves stability, convergence, and performance.

**Strengths:**

1. This work investigates an interesting aspect, the generalization of world models that learn the dynamics of the environment. Very limited work has been done in this facet of RL, thus it will share significant insights with the DRL research community.

2. The paper followed a structured methodology to analyze the world model and its representation errors. They interpret the learned model function as stochastic differential equations (SDEs) and model the variation as Brownian motions.

3. I liked the way they theoretically analyzed it case-by-case and established connections with prior findings.

4. The paper articulately presents the findings of zero-drift error as a regularizer and the Jacobian correction term for non-zero-drift representation error.  It systematically proves its hypotheses and shows evidence against the claims. They presented corresponding formulas and interpretations.

**Weaknesses:**

1. The paper is very thorough in terms of theoretical derivation. However, in my opinion, the experimental section of the paper is somewhat lacking. It utilizes only two tasks from Mujoco to prove the efficacy of the approach. More diverse tasks from other benchmarks and robust perturbations will certainly improve the paper.

2. The experimental evaluation is limited to reward comparison. However, it would be interesting to see some visualization of how the trajectories unfold in the case of both types of errors and with Jacobian regularization.

**Questions:**

1. It appears in different tables different values of $\lambda$ (the regularization weight in eq. 20) have been used. How sensitive the models are to the value of this hyperparameter? Do you have any suggested range for better performance?

2. Do you observe any substantial relation between $\lambda$ and the task horizon?

**Limitations:**

While the paper discusses the potential social impact of the work, it doesn’t discuss any limitations. I believe the characterization of the models as SDE and the use of Brownian motion as variation have certain contributions to the identified claims. Other interpretations may alter the findings.

---

> ### Author Rebuttal · Authors · 2024-08-07
>
> Thank you for your positive feedback and suggestions! Based on your valuable feedback, we have improved the empirical analysis of our proposed regularization schemes with some new experiments and visualizations.
>
>  ### **A1. Additional Experiments on Benchmark Environment**
>
> Thanks for the constructive suggestion. We intend to add 1-2 additional experiment environments to showcase robustness against perturbation brought by the Jacobian regularization in the final paper. Here we include a comparison of baseline method and Jacobian regularization in the challenging 2D environment -- Crafter [1] under noise perturbation. We add a Gaussian noise with 0 mean and 0.25 variance on the image state whose value ranges from -0.5 to 0.5 and apply masks with various percentages.
>
> |  mask $\beta \%, N(0, 0.25)$     | Baseline | Jac Reg ($\lambda= 0.1$) | Jac Reg ($\lambda = 0.01$) | Jac Reg ($\lambda = 0.001$) |
> |-------|---------|----------|-----------|---------|
> | $\beta=20\%$   | 9.1 | 16.96   | 17.14    | 23.18     |
> | $\beta=60\%$   | 8.51| 14.11   | 14.03    | 17.47     |
> | $\beta=100\%$  | 6.45 | 13.84   | 10.29    | 14.89     |
>
> We would like to include a note that within the very limited rebuttal timeframe, the baseline method finished fewer steps than its Jacobian counterparts, thus earlier checkpoints are used for baseline in this comparison. We observed that the baseline performance has already plateaued according to the train/eval curves, although its performance can still improve with more training steps. We will keep the baseline experiment running and update this table should there be any change in the performance.
>
> [1]: Danijar Hafner, 'Benchmarking the Spectrum of Agent Capabilities,' 2021
>
> ### **A2. Visualization of Reconstructed State Trajectories of Models Trained with/without Jacobian Regularization**
>
> Thanks for your great suggestions. We have added visualizations of reconstructed state trajectory samples in the revision to showcase the error propagation of exogenous zero-drift and non-zero drift error signals in latent states with and without Jacobian regularization. Please see Figure 1 & 2 in the attached PDF file.
>
> As shown in the figure, the reconstructed states for the baseline model without Jacobian regularization appear fuzy, indicating the model has not correctly captured the dynamics of the environment; whereas the reconstructed states for model with Jacobian regularization are sharp and correctly reflect the dynamics of the environment. The visual comparison highlights the robustness brought by Jacobian regularization against latent noises.
>
>
> ### **A3. Responses to Questions**
>
> ### Q1. On the effects of $\lambda$ on model’s generalization
>
> Thanks for raising the question. The choice of $\lambda$ can have subtle influence on the model's performance. The optimal value for $\lambda$ depends on the environment and the task. We suggest readers to try out $\lambda$ in the following ranges [0.1, 0.01, 0.001]. Please see table in A1, where we present crafter scores for 3 different values of $\lambda$ under a Gaussian noise with 0 mean and 0.25 variance on the input image states (where pixel values range from -0.5 to 0.5) with various masks.
>
> We will include a section to discuss $\lambda$'s influence for different environments in the final version of the paper.
>
>
> ### Q2. On the relation between $\lambda$ and task horizon
>
> We did not observe a substantial relationship between $\lambda$ and task horizon in our current experiments. However, we hypothesize that as task horizon gets longer, larger regularization weights ($\lambda$ around 0.1) would be more beneficial due to their tighter control on error propagation. Due to time limitations during the rebuttal period, we were unable to conduct these additional experiments, but we plan to study this further in future works.
>
>
> ### **A4. Response to Limitation**
>
>
> We appreciate you raising this point. In the revision, with more page space, we will include a separate section addressing the limitations of our work. We will highlight that the main limitation of the SDE interpretation of world models (WM) is its restriction to a popular family of world models for theoretical soundness. As mentioned in the paper (lines 123-125), "we consider a popular class of world models, including Dreamer and PlaNet, where $\{z, \tilde{z}, \tilde{s}\}$ have distributions parameterized by neural networks’ outputs, and are Gaussian when the outputs are known."
>
> In addition, our results on the approximation error of latent representation focus on CNN (and similar) models and future work is needed to generalize the study to models such as transformers. We are exploring this direction in our future work.

---

> > ### Author Response · Authors · 2024-08-11
> >
> > Thank you for your positive feedback. We have carefully addressed your comments in our responses.
> >
> > We kindly remind you to review our responses, and we are happy to address any further questions you may have.
> >
> > Thank you once again for your thoughtful insights and for contributing to the improvement of our work.

---

### Author Rebuttal · Authors · 2024-08-07

Dear reviewers,

We sincerely thank your comments and constructive suggestions. Below, we address the reviewer’s concerns point by point. We also attach a one-page PDF for visualizations and graphs.

---

### Decision · Program_Chairs · 2024-09-25

**Decision:**

Reject

**Comment:**

The paper takes an SDE-based perspective on understanding errors and generalization in learned latent dynamics models, and ultimately propose to regularize a learned dynamics model with the Jacobian w.r.t. the current state, leading to eq. (20). This adds generalization against unseen states and encoder errors on walker and quadruped tasks, and also seems to help improve convergence on a walker task. Generalization remains an active and open topic, and advancing our understanding of it is important to the community.

We thoroughly discussed this paper and came to the conclusion that there are still a few remaining concerns that limit publication in the current form. These need another reviewing cycle to address:

1. The most significant concern, raised by reviewers pwHm and waKG, is that the experimental results are not extensive and do not evaluate on any established generalization setting. Despite connecting to the models used the Dreamer and TD-MPC variants, the experimental results are only on walker and quadruped systems with artificially created noisy states. Reviewer waKG raised the concern that there are no comparisons to works that perform image augmentations, and the authors responded with an experiment in the walker environment. It would have been convincing to compare to the standard Dreamer/TD-MPC models trained on the more complex environments, and also to use established benchmarks for robustness/generalization, e.g., as discussed in this next point:
2. Generalization in world models (and MDPs) can go significantly beyond the formulation they consider, e.g., as described in A Survey of Zero-shot Generalisation in Deep Reinforcement Learning. That paper also summarizes a number of existing generalization benchmarks (e.g., the distracting control suite) and methods that experimentally this paper could have compared with. However, none of them are used (or referenced). For this reason, the paper seems too disconnected from the existing research on generalization in RL --- neither experimentally nor their related work/other discussions discuss other literature of generalization in RL. This makes the claim of  "unraveling and improving generalization" unsupported.
3. The idea of a Jacobian regularization is straightforward and can be added without going through the SDE formulation or bounds. There could be some theoretical interest and contribution from the SDE bounds and relating them to generalization and the Jacobian, but they are also little disconnected from the experimental setting because the Jacobian regularization could also be interpreted from other perspectives without needing the SDE bounds. While the latent dynamics SDE with an error model is interesting to study, and bound against the reference 0-error SDE, we feel the claims to generalizing to non-trivial modeling errors are not well-supported. We do not find the additional theory in Section 3 to be experimentally supported.
4. We assume the finalized Crafter results will at least be consistent with Table 2, where they have a similar experiment for the walker and quadruped environments. While it would improve the generalization, it is not a significant kind of generalization as the perturbations they use in the environment are synthetic and not surprising given the gradient penalty they added for training the model. Furthermore, the table with Crafter results they posted seems a little misleading as 1) it is unlikely that they would surpass the performance of the baseline method in the original, unnoised environment (but it would be a positive result if they do!), and 2) it only uses a single seed (actually, all of the experiments seem to use a single seed --- we do assume the main trends would continue holding over multiple seeds, but it is scientifically important to show this, especially if they are comparing on the standard benchmarks where other methods are reporting averaged performance)